# Unveiling the role of local metabolic constraints on the structure and activity of spiking neural networks

Ismael Jaras[1,2]*, Marcos E. Orchard[1☉], Pedro E. Maldonado[2,3☉], Rodrigo C. Vergara[3,4]*

1 Department of Electrical Engineering, Faculty of Mathematical and Physical Sciences, University of Chile, Santiago, Chile, 2 Neurosystems Laboratory, Department of Neuroscience, Faculty of Medicine, University of Chile, Santiago, Chile, 3 Centro Nacional de Inteligencia Artificial CENIA, Santiago, Chile, 4 Departamento de Kinesiología, Facultad de Artes y Educación Física, Universidad Metropolitana de Ciencias de la Educación, Santiago, Chile

☉ These authors contributed equally to this work.
* ismael.jaras@ing.uchile.cl (IJ); rodrigo.vergara_o@umce.cl (RCV)

**Data availability statement:** The source code associated with this study is available on

## Abstract

Understanding the intricate interplay between neural dynamics and metabolic constraints is crucial for unraveling the mysteries of the brain. Despite the significance of this relationship, specific details concerning the impact of metabolism on neuronal dynamics and neural network architecture remain elusive, creating a notable gap in the existing literature. This study employs an energy-dependent neuron and plasticity model to analyze the role of local metabolic constraints in shaping both the dynamics and structure of Spiking Neural Networks (SNN). Specifically, an energy-dependent version of the leaky integrate-and-fire model is utilized, along with a three-factor learning rule that incorporates postsynaptic available energy as the third factor. These models allow for fine-tuning sensitivity in the presence of energy imbalances. Analytical expressions predicting the network's activity and structure are derived, and a fixed point analysis reveals the emergence of attractor states characterized by neuronal and synaptic sensitivity to energy imbalances. Analytical findings are validated through numerical simulations using an excitatory-inhibitory network. Furthermore, these simulations enable the study of SNN activity and structure under conditions simulating metabolic impairment. In conclusion, by employing energy-dependent models with adjustable sensitivity to energy imbalances, our study advances the understanding of how metabolic constraints shape SNN dynamics and structure. Moreover, in light of compelling evidence linking neuronal metabolic impairment to neurodegenerative diseases, the incorporation of local metabolic constraints into the investigation of neuronal network structure and activity opens an intriguing avenue for inspiring the development of therapeutic interventions.

## Author summary

Our study explores the complex relationship between brain activity and energy use, an area crucial for understanding how the brain functions. Despite its importance, the

GitHub. Researchers and readers can access, download, and collaborate using the provided code to replicate or extend our analyses. The specific repository link is as follows: https://github.com/Wiss/edsnn.

**Funding:** This work was partially supported by the National Center for Artificial Intelligence CENIA, FB210017, BASAL, ANID, awarded to P.E.M. and R.C.V., by FONDECYT Grant 1250036, and by the Advanced Center for Electrical and Electronic Engineering, BASAL Project AFB240002, awarded to M.E.O. The authors also acknowledge support from ANID-PFCHA/Doctorado Nacional/2019-21190330 for funding Ismael Jaras's doctoral studies.

**Competing interests:** The authors have declared that no competing interests exist.

impact of energy availability on brain cell dynamics and network structure is not well understood. To address this, we developed a model that integrates energy constraints into the behavior of neurons and synapses. Using an energy-dependent neuron model and a synaptic plasticity rule that considers energy availability, we analyzed how these constraints affect the brain's activity and structure. Our model allows us to adjust sensitivity to energy imbalances, offering detailed predictions about network behavior. Through analytical and numerical methods, we discovered that energy constraints can lead to distinct stable states in neural networks. We also simulated scenarios of metabolic impairment, providing insights into how energy deficiencies might affect brain function. This work enhances our understanding of the role of metabolism in brain activity and suggests new directions for research into neurodegenerative diseases, where metabolism is often impaired.

## Introduction

Neural tissue consumes a higher amount of energy resources compared to other somatic tissues [1], where electric activity corresponds up to 75% of neuron expenses [2,3]. This amount of resource consumption is not trivial as neurons are extremely sensitive to resource deprivation, particularly to decrease of oxygen and glucose levels on blood or cerebrospinal fluid (CSF) [4]. Furthermore, many neurodegenerative diseases, such as Parkinson's disease, Leigh, amyotrophic lateral sclerosis (ALS), and Alzheimer's disease have been proposed to occur due to energy management impairment [5–10]. Despite the recognized relevance of energy management in the nervous system [11,12], energy management is rarely considered as a relevant factor in neuroscience and in neural modeling.

Energy management also appears relevant in common behavioral activities. Neural activity elicits local increases in blood flow (neurovascular coupling), glucose uptake, and oxygen consumption [13], being this same energy management mechanism the one used in fMRI to map behavior into the brain [14]. As such, fMRI evidence could also be interpreted as a correlate of behavior with energy management in neural tissue. However, the energy management of active neurons is even more finely tuned. For instance, neurons outsource glycolytic activity to astrocytes leaving energy levels fairly constant during time [15]. Besides the energy substrates given by astrocytes during neuronal activity, neurons are provided with intrinsic mechanisms that buffer ATP levels. Accordingly, neuron mitochondria raise ATP synthesis in response to an increment in synaptic stimuli [10,16–19]. Together, these arguments present robust evidence that supports the importance of energy administration for maintaining the brain's proper function, thereby leading to the development of modeling strategies that account for metabolic dynamics.

Attempting to establish the significance of energy management as a pivotal factor influencing neural systems, the Energy Homeostasis Principle (EHP) was introduced in [11,12]. The rationale behind this principle posits that neurons, in meeting their metabolic needs, inadvertently address behavioral challenges as an epiphenomenon. EHP allows the reinterpretation of several physiological mechanisms. One of them is the synapses, which is reinterpreted as an energetic control mechanism. While neurons lack direct control over presynaptic stimulation, they exert influence over the "weight" or strength of such activity. Therefore, synaptic weights can be used to regulate incoming energy expenses from previous neurons. Critically, synaptic weight modifications, imply changes in neural network dynamics and ultimately behavior.

Consequently, the core hypothesis derived from the EHP posits that energy management significantly shapes synaptic weights, consequently impacting the broader dynamics of neural systems.

While the Energy Homeostasis Principle (EHP) has made strides in addressing certain gaps in understanding, a comprehensive grasp of the effects of local energy constraints at a network level remains elusive. To advance our comprehension of the impact of metabolic constraints on the brain, an effective strategy involves formalizing neuronal and synaptic models. This formalization should encompass experimentally observed energy dependencies, affording the opportunity to analytically scrutinize the implications of metabolic constraints as well as simulate their effects.

This work fills the gap by committing to the previous approach. Here we create an energy-dependent Spike-timing-dependent plasticity rule and, jointly with a previously introduced energy-dependent neuronal model [20], we study the role of local metabolic constraints on the structure and activity of spiking neural networks, particularly in Excitatory-Inhibitory (E-I) networks. Our exploration employs two complementary methods to assess the impact of metabolic constraints at the network level. Initially, we conduct a mathematical analysis to understand the consequences of incorporating energy dependencies into both neurons and synapses. Subsequently, we undertake simulations to further analyze the effects of metabolic constraints, providing a basis for comparison against the analytical results.

Through our mathematical analysis, we find the conditions under which the network converges toward a fixed point. Then, we compare theoretical predictions with numerical simulations, observing a good agreement between both. Regarding different simulation cases, we divide the experiments into three main scenarios. Firstly, we study the dynamics and structure of the E-I networks focusing solely on synaptic sensitivity to energy imbalances. Following this, we extend our simulations to include neuronal sensitivity to energy imbalances. Finally, motivated by the evidence linking neurodegenerative diseases to metabolic impairments, we investigate the effect of neuronal impaired metabolic productions on the dynamics and structure of the network.

This study aims to advance our understanding of how local metabolic constraints influence the structure and activity of spiking neural networks. To achieve this, we introduce a novel energy-dependent plasticity rule and investigate the impact of local metabolic constraints through both analytical methods and numerical simulations. In doing so, we have focused on simulating *in vitro* neural network conditions, as those experimental settings are the most accessible to explore the interrelation between metabolism and plasticity.

## Materials and methods

### Energy dependent leaky integrate-and-fire (EDLIF)

The Leaky integrate-and-fire (LIF) model is a single-compartment model describing the sub-threshold dynamics of the neuron membrane potential. Despite its simplicity, the LIF model it is widely used for modeling neuronal dynamics. The LIF model does not take into account neuronal energetics affecting his dynamics, however, ultimately, the function of the neuron rests on its ability to balance its energy expenditure and production. Within this balance, the sodium-potassium pump is a key actor, considering that it relies on available ATP to restore the resting potential allowing action potentials to occur [21]. Consequently, neuronal behavior depends on the adequate balance between energy production and expenditure. The main objective of the EDLIF [20] model is to include the energy dependence in the neuronal dynamics while maintaining the LIF model's simplicity. To accomplish that, the EDLIF model

makes the neuronal behavior explicitly dependent on the available neuronal energy through the energy-dependent partial repolarization mechanism.

As in the LIF model, in the EDLIF model the membrane's voltage of the neuron is described as follows:

$$C_m \dot{v}(t) = -g_{leak}(v(t) - v_{rest}) + I(t), \tag{1}$$

where $C_m$ is the membrane capacitance, $v_{rest}$ the equilibrium potential of the leak channel, $g_{leak}$ is the conductance associated with the current leakage, and $I(t)$ the input current. Unlike the LIF model, in the EDLIF model the reset potential after an action potential occurs is metabolically dependent. This dependency is achieved by the inclusion of a partial repolarization mechanism [22], which works by resetting the potential of the capacitor to $v_{reset} = \beta v_{th}$ after an action potential occurs, where $v_{th}$ is the firing threshold and $\beta$ is called the reset parameter. In the EDLIF model, the reset parameter ($\beta$) depends on ATP as follows:

$$\beta(A(t)) = 1 + \rho \left[ 2 - \frac{2}{1 + e^{-\frac{A_H - A(t)}{A_H} \gamma}} \right], \tag{2}$$

where $\rho = v_{rest}/v_{th} - 1$, $A_H$ is the homeostatic ATP level, and $A(t)$ the neuron's ATP level. This relationship represents the membrane repolarization voltage values after an action potential occurs as a function of the available ATP in the neuron. Given that the precise curve of the repolarization membrane voltage depending on ATP is unknown, a sensitivity parameter $\gamma$ is introduced. This parameter provides the flexibility to adjust the intensity at which the repolarization membrane voltage value is affected by ATP level changes.

When a constant suprathreshold stimuli ($I_{inj} = I_0 > v_{th} g_{leak}$) is applied to the neuron's model with initial condition $v_m(t = 0) = v_{reset}(A)$, the membrane's potential satisfies the following trajectory:

$$v_m(t) = v_\infty \left[ 1 - e^{-t/\tau_m} \right] + v_{reset}(A) e^{-t/\tau_m}, \tag{3}$$

where $v_\infty = (I_0/g_{leak} + v_{rest})$. The time it takes for a spike to be generated (i.e., the interspike interval, $s_0$) is obtained by imposing $v_m(s_0) = v_{th}$ [23]:

$$s_0(A) = \tau_m \ln \left[ \frac{v_\infty - v_{reset}(A)}{v_\infty - v_{th}} \right]. \tag{4}$$

Therefore, the corresponding ATP-dependent spiking-rate $\nu(A)$ for constant available energy in the neuron is described by the following equation:

$$\nu(A) = (\tau_{ref} + s_0(A))^{-1}, \tag{5}$$

where $\tau_{ref}$ is a time constant accounting for neuronal refractoriness. In our brains, the available energy in neurons changes through time, so it is necessary to include this time-dependence in the available energy to estimate the firing rate of the neuron. Fig 1 shows how the neuron's firing rate response to an injected constant current changes when the neuronal sensitivity to energy imbalances ($\gamma$) is modified and the available energy changes through time.

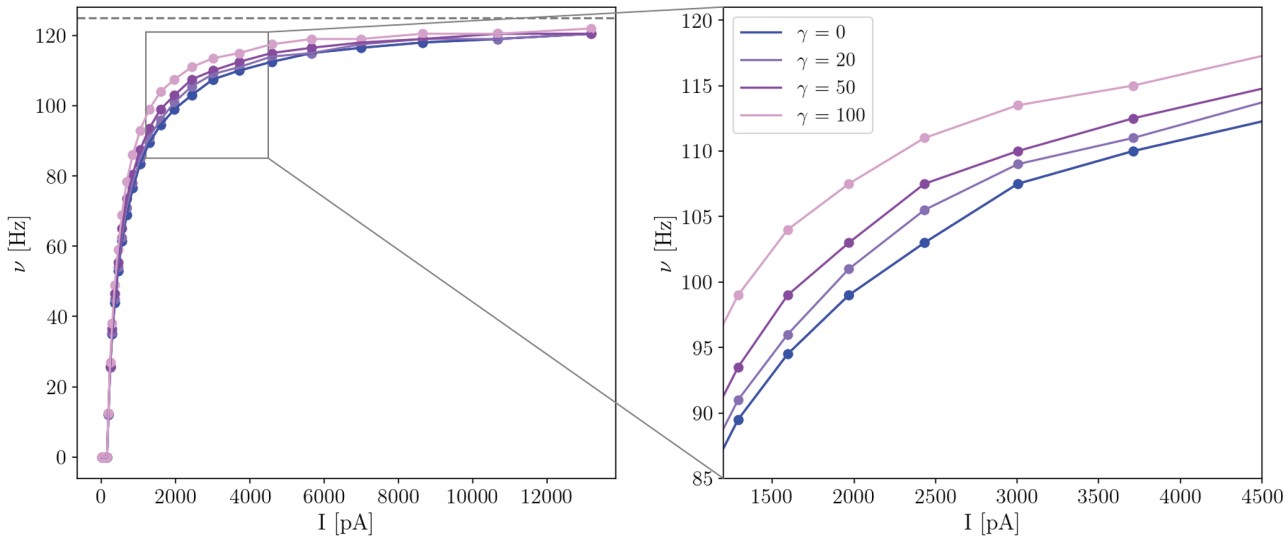

**Fig 1. Current-frequency mapping for neurons with different $\gamma$ sensitivity.** The current-frequency mapping depends on the neuronal sensitivity to energy imbalances. If $\gamma$ is higher the neuron's firing rate saturates before (with a smaller current) in contrast to a neuron with lower neuronal sensitivity to energy imbalances.

**Neuronal ATP dynamics.** In the EDLIF model, the ATP dynamics are characterized by considering two processes: those that supply ATP to the neuron ($A_s(t)$), and those that consume ATP ($C(t)$). The available ATP in the neuron ($A(t)$) can be formalized as follows [11]:

$$\dot{A}(t) = A_s(t) - C(t), \tag{6}$$

and the neuron's homeostatic mechanisms attempt to keep ATP levels close to the homeostatic ATP level ($A_H$). The ATP production dynamics is mathematically formalized by Eq 7. Thus, ATP production depends on the actual ATP level with respect to a homeostatic one ($K(A_H - A(t))$) and a basal production term ($A_B$) accounting for resting potential and housekeeping activities:

$$A_s(t) = K(A_H - A(t)) + A_B, \tag{7}$$

where the parameter $K$ (($1/ms$) units) is the rate at which ATP is produced.

Regarding energy consumption, we consider four energy-consuming activities in this study. Namely, the energy related to the neuron's housekeeping activities $E_{hk}$, the energy associated with maintaining the resting potential $E_{rp}$, the total energy expended by the neuron's action potentials $E_{ap}$, and the total energy associated with receiving postsynaptic potentials due to a presynaptic neighbor neurons having action potentials ($E_{syn}$). Both rate consumptions $E_{hk}$ and $E_{rp}$ are measured in energy per time units. On the other hand, both consumption activities $E_{ap}$ and $E_{syn}$ refer to the total amount of energy consumed by those activities. To obtain the corresponding rate consumption for $E_{ap}$ and $E_{syn}$ we use temporal kernels. In this manner, the amount of energy expended per time unit associated with action potentials (or postsynaptic potentials) is described as $C_{ap}(t) = E_{ap}\varepsilon_{ap}(t)$ (or $C_{syn}(t) = E_{syn}\varepsilon_{syn}(t)$), where $\varepsilon_x(\cdot)$ is the kernel associated to the process $x$ and $\int_0^\infty \varepsilon_x(t)dt = 1$.

Originally, in the EDLIF model the production and consumption of ATP is measured in $mM/ms$, while the available ATP $A(t)$ is measured in $mM$. However, experimentally, it is difficult to precisely measure ATP concentrations in single neurons or populations. To overcome this difficulty and to define a more general framework, here we will use percentage units to quantify available energy ($A(t)$), homeostatic energy ($A_H = 100\%$), and activity-related energy consumption such as action potential energy consumption ($E_{ap}$), house-keeping ($E_{hk}$), resting potential ($E_{rp}$), and postsynaptic energy consumption ($E_{syn}$).

**Static synapses and energy consumption in the neuron.** To understand in detail the energy perturbations induced by presynaptic neurons in postsynaptic energy levels through synapses, let us analyze the neuron's energy dynamics when several presynaptic neurons are connected to it. In a network, each neuron's energy evolves following Eq 6, where $A_s = K(A_H - A) + A_B$, and the energy consumption $C$ can be divided into the following terms:

$$C(t) = C_B + C_{ap}(t) + C_{syn}(t), \tag{8}$$

where $C_B$ represents the basal energy consumption associated with maintaining the resting potential and general housekeeping activities (which is equal to the basal production $A_B$ in this work). $C_{ap}$ is the energy consumption accounting for the neuron action potentials, and $C_{syn}$ is the energy consumption related to receiving action potentials from other neurons through the synapses. Please realize that $C_{ap}(t)$ ($C_{syn}(t)$) is different than $E_{ap}$ ($E_{syn}$). The former is the action potential (postsynaptic) energy consumption through time, and the latter is the total amount of energy related to that activity. Thus, $C_{ap}$ ($C_{syn}$) is obtained using $E_{ap}$ ($E_{syn}$) and a kernel function $\varepsilon(\cdot)$, which describes how the total energy consumption is expended through time.

The solution to Eq 6 with an arbitrary energy consumption $C(t)$ is described by Eq 9:

$$A(t) = A_H - \int_0^t e^{-(t-t')/\tau_A} C(t')dt', \tag{9}$$

where $K = 1/\tau_A$ is the rate at which ATP can be produced in the neuron (see Eq 7).

For one presynaptic action potential propagating through a synapse with fixed strength $w$, the synaptic consumption on the postsynaptic neuron can be described by:

$$C_{syn}(t) = E_{syn}\tilde{w}\Theta(t - t_s)\varepsilon_{syn}(t - t_s), \tag{10}$$

where $\Theta(\cdot)$ denotes the Heaviside step function, $E_{syn}$ is the total energy expended for an incoming presynaptic action potential, $\tilde{w}$ is the absolute value of the normalized synaptic strength ($\tilde{w} = |w/w_{max}|$, where $w_{max}$ is the maximum allowed weight strength) and $\varepsilon_{syn}(t)$ is a kernel describing how the synaptic energy expenditure is expended through time ($\int_0^\infty \varepsilon_{syn}(t)dt = 1$). If we choose a kernel that can be linearly summed up, then the synaptic energy expenditure in the postsynaptic neuron due to several presynaptic cells with their respective spike times is:

$$C_{syn}(t) = E_{syn} \sum_{\text{synapse } k} \sum_{\text{spike } s} \tilde{w}_k\Theta(t - t_s)\varepsilon_{syn}(t - t_s). \tag{11}$$

Similarly, for each action potential in the postsynaptic neuron, the energy expenditure can be described by:

$$C_{ap}(t) = E_{ap}\Theta(t - t_s)\varepsilon_{ap}(t - t_s), \tag{12}$$

where $E_{ap}$ is the energy expenditure related to one action potential and $\varepsilon_{ap}$ is analogous to $\varepsilon_{syn}$, but for the postsynaptic action potentials ($\int_0^\infty \varepsilon_{ap}(t)dt = 1$). If $\varepsilon_{ap}$ can be linearly summed up, then Eq 12 reads:

$$C_{ap}(t) = E_{ap} \sum_{\text{spike } s} \Theta(t - t_s)\varepsilon_{ap}(t - t_s). \tag{13}$$

To find closed-form expressions for the available energy in the postsynaptic neuron, it is possible to define $\varepsilon_{ap}$ and $\varepsilon_{syn}$ kernels and plugin Eqs 11 and 13 into Eq 6. In particular, if exponential kernels are used for $\varepsilon_{ap}$ and $\varepsilon_{syn}$, Eq 11 reads:

$$C_{syn}(t) = \sum_{\text{synapse } k} \sum_{\text{spike } f} \frac{\tilde{w}_k E_{syn}^k}{\tau_{syn}^{A,k}} \Theta(t - t_f)e^{-(t-t_f)/\tau_{syn}^{A,k}}, \tag{14}$$

where $\tau_{syn}^{A,k}$ is the energy time constant of the $k$th synapse and describes how *fast* is the change in the postsynaptic neuron's energy given the incoming action potential through the $k$th synapse. Note that if $w_k = w_{max}$ the total energy consumption in the postsynaptic neuron produced by the presynaptic neuron sending one action potential through synapse $k$th is $E_{syn}^k$. Thus, Eq 14 guarantees that for the maximum allowed synaptic weight, the total energy consumption perceived by the postsynaptic neuron is exactly $E_{syn}^k$ and a fraction ($|w_k/w_{max}|$) of $E_{syn}^k$ otherwise. Similarly, using an exponential kernel, Eq 13 can be expressed as:

$$C_{ap}(t) = \Theta(t - t_s)\frac{E_{ap}}{\tau_{ap}}e^{-(t-t_s)/\tau_{ap}}, \tag{15}$$

where $\tau_{ap}$ is the time constant defining how *fast* is $E_{ap}$ expended through time. Eq 15 guaranties that the neuron will consume a total amount of $E_{ap}$ for each action potential.

Therefore, plugin Eqs 15 and 14 into Eq 9, we have a closed-form expression for the neuron's energy dynamics under exponential kernels ($\varepsilon_{ap}$ and $\varepsilon_{syn}$) for energy consumption:

$$A(t) = A_H - \overbrace{\frac{E_{ap}\tau_A}{\tau_A - \tau_{ap}} \sum_{\text{spike } s} \left[e^{-(t-t_s)/\tau_A} - e^{-(t-t_s)/\tau_{ap}}\right]}^{C_{ap}(t) \text{ contribution}}$$
$$- \underbrace{\sum_{\text{synapse } k} \sum_{\text{spike } f} \frac{E_{syn}^k \tilde{w}_k \tau_A}{(\tau_A - \tau_{syn}^{A,k})} \left[e^{-(t-t_f)/\tau_A} - e^{-(t-t_f)/\tau_{syn}^{A,k}}\right]}_{C_{syn}(t) \text{ contribution}}. \tag{16}$$

Eq 16 describes the energy dynamics of a single neuron within a network, given all incoming synapses strengths and spike-times. Thus, it allows to precisely quantify and analyze the energy dynamics of each neuron within a network with arbitrary architecture.

### Energy dependent Spike-timing-dependent plasticity

We aim to investigate how local energy limitations influence network attributes. While having an energy-dependent single-neuron model is crucial, it's not the only consideration. We must also factor in the metabolic impact on synapses. The energy administration of the neuron is essential for its adequate operation, and considering the significant effect of these restrictions on synaptic activity [9,10,24,25], the need to have synaptic transmission rules that depend on the energy management of the cell naturally emerges.

**Energy dependent Spike-timing-dependent plasticity model.** Here we leverage previously defined plasticity models to introduce an energy-dependent synaptic plasticity rule.

Particularly, we introduced an energy-dependent plasticity rule for synaptic strength. We follow the formalization of STDP introduced in [26]:

$$\Delta w = \begin{cases} \lambda f_+(w)\kappa(\Delta t), & \text{if } \Delta t > 0. \\ -\lambda f_-(w)\kappa(\Delta t), & \text{if } \Delta t \le 0, \end{cases} \tag{17}$$

where the synaptic efficacy $w$ is normalized to [0,1]. The learning rate $\lambda$ ($0 < \lambda \ll 1$) scales the magnitude of the individual weight change, $\kappa(\cdot)$ is a temporal filter $\kappa(x) = e^{-\|x\|/\tau_{stdp}}$, and $f_{\pm}(\cdot)$ is defined as follows:

$$f_+(w) = (1-w)^{\mu_+} \text{ and } f_-(w) = \alpha w^{\mu_-}, \tag{18}$$

with $\alpha > 0$ accounting for the asymmetry between the scales of potentiation and depression.

In general, neurons harbor energetic/metabolic sensors such as AMP-activated protein kinase, which tends to restore ATP concentration by decreasing energy consumption while increasing energy production after different energetic challenges [27–29]. This same mechanism has been linked to plasticity [29], where they were able to show that energetic stress can suppress LTP, while treatments targeting energetic pathways can rescue it by removing the energetic stress. This findings inspired our LTP energy-dependent plasticity rule. Specifically, when glycolysis is pharmacologically inhibited (thus ATP production from glucose is inhibited), LTP is suppressed [29]. Therefore, our energy-dependent STDP rule should suppress LTP when there is a low energy level. Following this observation, we proceed to extend Eq 17 to account for energetics, while keeping in mind that this is a practical simplification of a more complex biological phenomena, but which allows deepening the understanding of metabolism and plasticity interactions, and its impact on neural network dynamics.

To account for energetics in STDP, it is possible to extend $f_+(w)$ in Eq 18 to include the energy level $A$ of the postsynaptic neuron, obtaining $f_+(w, A)$. There are several ways to mathematically formalize the inclusion of postsynaptic energy level in $f_+(w)$, but we follow a similar rationale as the one used in EDLIF neuron, namely, we want energy-dependent STDP to be *sensitive* to energy imbalance. Hence, we can investigate not only how energy affects STPD, but also how energy imbalance affects STDP given a certain level of *energy imbalance sensitivity* in that synapse's plasticity. Using the rationale mentioned above we extend $f_+(w)$ to $f_+(w, A)$ as follows:

$$f_+(w, A) = f_+(w) e^{-\eta \frac{A_H - A}{A_H}}, \tag{19}$$

where $\eta$ is the sensitivity of the synapse's plasticity to energy imbalances, $A_H$ is the homeostatic energy level, and $A$ is the postsynaptic energy level. STDP effects can be thought as a change in postsynaptic receptor density. Given that the postsynaptic neuron has to modify its receptor's density, their energy level is affecting receptor density modifications. Please note that if the synapse is not sensitive to energy imbalances (*i.e.*, $\eta = 0$), then we recover the original STDP rule.

The postsynaptic energy level can be interpreted as a plasticity moderator. Thus, our energy-dependent STDP rule is a special case of a three-factor learning rule, with postsynaptic available energy $A$ as modulator.

For the depression updating function $f_-(w)$ we keep the original formulation ($f_-(w) = \alpha w^{\mu_-}$, see Eq 18) because we don't have conclusive evidence justifying the depression's dependence on postsynaptic available energy. However, energy may be also affecting depression.

Plugging Eq 19 into Eq 17, it is possible to describe weight's update accounting for pre- and postsynaptic spike-time and postsynaptic energy level:

$$\Delta w = \begin{cases} \lambda f_+(w, A) e^{-\Delta t/\tau_+}, & \text{if } \Delta t > 0. \\ -\lambda f_-(w) e^{\Delta t/\tau_-}, & \text{if } \Delta t \leq 0. \end{cases} \tag{20}$$

In principle, Eq 20 allows to include weight $w$ dependencies in the updating rule if $\mu_\pm > 0$. In those cases, the updating rule is called a multiplicative rule, while if $\mu_\pm = 0$ the updating rule is additive, because weight update is independent of the current weight value. Multiplicative rules generate uni-modal weight distributions, while additive rules generate bi-modal weight distributions [26]. Bi-modal weights distributions are observed in biology, and because the mathematical tractability of STDP additive rules is easier than multiplicative ones, in what follows we decided to focus on the additive update rule for weights update ($\mu_\pm = 0$). However, there is also significant biological evidence supporting multiplicative STDP as well. For this reason, we define Eq 20 in a general form, allowing the exploration of multiplicative energy-dependent STDP in future works.

## Network model

There is biological evidence suggesting the existence of excitatory-inhibitory (E-I) balanced networks in different parts of the brain, such as the CA3 region of the hippocampus, basal ganglia, and the primary visual cortex [30–33]. Thus, E-I networks appear as a suitable model to explore neural networks' dynamics and structure. However, in general, energy or metabolic constraints are not considered when simulating and studying E-I networks.

To mathematically describe an E-I network with $n_E$ excitatory and $n_I$ inhibitory neurons, we introduce the network's weight matrix $\mathbf{w}$ of $(n_E + n_I) \times (n_E + n_I)$ dimension:

$$\mathbf{w} = \left( \begin{array}{c|c} \mathbf{w^{EE}} & \mathbf{w^{EI}} \\ \hline \mathbf{w^{IE}} & \mathbf{w^{II}} \end{array} \right),$$

where the $i, j$ element of $\mathbf{w}$ correspond to the synapse from the $j$th presynaptic neuron to the $i$th postsynaptic neuron. $\mathbf{w^{EE}}$ is the $n_E \times n_E$ matrix containing all the excitatory-excitatory connections, $\mathbf{w^{IE}}$ is the $n_I \times n_E$ matrix containing all the excitatory-inhibitory connections, $\mathbf{w^{EI}}$ is the $n_E \times n_I$ matrix containing all the inhibitory-excitatory connections, and $\mathbf{w^{II}}$ is the $n_I \times n_I$ matrix containing all the inhibitory-inhibitory connections. Because both LTP and LTD depend on the activation of NMDA receptors and are absent when the postsynaptic neurons are GABAergic [34], and that LTP and LTD mechanisms in inhibitory neurons are widely diverse in their physiology [35], only excitatory-excitatory connections ($\mathbf{w^{EE}}$) are plastic in our E-I network. The main reason for this is that we cannot generalize the energy dependence observed for excitatory neurons, given that inhibitory neurons have relevant physiological differences compared to excitatory neurons.

The mean firing rate of each neuron in the $[t, t+T]$ interval is described by the $(n_E + n_I) \times 1$ vector $\bar{\boldsymbol{\nu}}_t$:

$$\bar{\boldsymbol{\nu}}_t = \left( \frac{\bar{\boldsymbol{\nu}}_t^E}{\bar{\boldsymbol{\nu}}_t^I} \right),$$

where $\bar{\boldsymbol{\nu}}_t^E$ is the $n_E \times 1$ dimension vector containing the excitatory mean firing rate, whereas $\bar{\boldsymbol{\nu}}_t^I$ contains the inhibitory mean firing rates in a $n_I \times 1$ dimension vector.

**Network's firing-rate.** To improve our analytical understanding of the E-I network, it is practical to have an approximation of the network's firing rate $\boldsymbol{\nu}$. In principle, using the LIF model (or the EDLIF model with no energy sensitivity to energy imbalances, *i.e.*, $\gamma = 0$) and knowing the model's parameters, the firing rate of a neuron excited by a constant current can be easily calculated. In a more general way, the firing rate of a neuron can be calculated by defining a function $\phi : \mathbb{R} \to \mathbb{R}$, mapping stimulation current to firing rate. If the neuron is insensitive to energy imbalance (*i.e., $\gamma = 0$*) and is excited with a supra-threshold current $I$, then the function $\phi(\cdot)$ can be found utilizing Eq 21:

$$T_n = \tau_{ref} + \tau_m \ln\left(\frac{v_\infty - v_{reset}}{v_\infty - v_{th}}\right), \tag{21}$$

where $\tau_{ref}$ is the neuron's refractory period, $\tau_m = C_m/g_{leak}$ is the neuron's time constant, and $v_\infty = I/g_{leak} + v_{reset}$. In this case, $v = T_n^{-1} = \phi(I)$, where $I$ is the constant incoming current to the neuron. Thus, the firing rate of each neuron under a constant current stimulation can be calculated knowing the neuron's parameters as well as the incoming constant current and the $\phi(\cdot)$ function, which is monotonically increasing in $I$. Therefore, if we can approximate the mean incoming current $\bar{I}_t$ to a neuron in a small time-window, it is possible to approximate the neuron's firing rate $\bar{\boldsymbol{\nu}}_t \approx \phi(\bar{I}_t)$, where $\bar{x}_t = T^{-1} \int_t^{t+T} x(t')dt'$. Hence, to approximate each neuron's firing rate, let us approximate the mean incoming current $\bar{I}_t$ to each neuron:

$$\begin{aligned} \bar{I}_t &= \frac{1}{T} \int_t^{t+T} I(t')dt' \\ &= \frac{1}{T} \int_t^{t+T} I_{\text{stim}} + \sum_{\text{synapse } k} \sum_{\text{spike } s} w_k \varepsilon_k(t' - t_s)dt'. \end{aligned} \tag{22}$$

Therefore, the mean incoming current to a neuron is the mean incoming synaptic current, plus the constant external incoming stimulation current $I_{\text{stim}}$ injected into the neuron. Eq 22 describes the mean incoming current to a neuron utilizing an $\varepsilon_k(\cdot)$ interaction kernel. If we use a delta Dirac interaction kernel (*i.e.*, $\varepsilon_k(\cdot) = \delta(\cdot)$), Eq 22 reads:

$$\begin{aligned} \bar{I}_t &= I_{\text{stim}} + \sum_{\text{synapse } k} w_k \underbrace{\frac{1}{T} \int_t^{t+T} \sum_{\text{spike } s} \delta(t' - t_s)dt'}_{\bar{\boldsymbol{\nu}}_t^k} \\ &= I_{\text{stim}} + \sum_{\text{synapse } k} w_k \bar{\boldsymbol{\nu}}_t^k. \end{aligned} \tag{23}$$

If we define the constant incoming stimulation current to each neuron as the $(n_E + n_I) \times 1$ dimension vector $\boldsymbol{I}_{\text{stim}} = \left[(\boldsymbol{I}_{\text{stim}}^E)^\top, (\boldsymbol{I}_{\text{stim}}^I)^\top\right]^\top$, and the mean incoming synaptic current as the $(n_E + n_I) \times 1$ dimension vector $\bar{\boldsymbol{I}}_t = \left[(\bar{\boldsymbol{I}}_t^E)^\top, (\bar{\boldsymbol{I}}_t^I)^\top\right]^\top$, Eq 23 can be expressed in matrix form as follows:

$$\bar{\boldsymbol{I}}_t = \boldsymbol{I}_{\text{stim}} + \overline{\text{w}}_t \bar{\boldsymbol{\nu}}_t. \tag{24}$$

It is possible to apply $\phi(\cdot)$ point-wise to obtain the approximated network's firing rate:

$$\begin{aligned} \bar{\boldsymbol{\nu}}_t &\approx \phi(\bar{\boldsymbol{I}}_t) \\ &= \phi(\boldsymbol{I}_{\text{stim}} + \overline{\text{w}}_t \bar{\boldsymbol{\nu}}_t). \end{aligned} \tag{25}$$

Eq 25 imposes a current-frequency constraint on the network. In general, $\phi(\cdot)$ defines a nonlinear relationship between incoming current and the neuron's firing rate. However, we can Taylor expand Eq 25 around $\boldsymbol{I}_{\text{stim}}$:

$$\overline{\boldsymbol{\nu}}_t \approx \phi(\boldsymbol{I}_{\text{stim}}) + \phi'(\boldsymbol{I}_{\text{stim}})\overline{\mathbf{w}}_t\overline{\boldsymbol{\nu}}_t$$
$$\Rightarrow \overline{\boldsymbol{\nu}}_t \approx [\mathbb{I} - \overline{\mathbf{w}}_t\phi'(\boldsymbol{I}_{\text{stim}})]^{-1}\phi(\boldsymbol{I}_{\text{stim}}), \tag{26}$$

where $\mathbb{I}$ is the identity matrix. Moreover, if $\phi(\cdot)$ is a linear transformation such as $\phi(x) = \xi x$, the following relationship holds:

$$\overline{\boldsymbol{\nu}}_t \approx [\mathbb{I} - \overline{\mathbf{w}}_t\xi]^{-1}\xi\boldsymbol{I}_{\text{stim}}. \tag{27}$$

Eqs 26 and 27 shows that, if the weights are static on average ($\Delta\overline{\mathbf{w}}_t = 0$), then the firing rate of the network converges towards an equilibrium. This observation is important because it highlights the fact that, in general, constant weights are needed to achieve a constant firing rate.

Eq 25 defines a relation determining plausible weight-frequency values, which in principle, are independent of the network's energy dynamics. In the Results section, we will explore how these constraints interact with the metabolic constraints at the network level.

**Simulation details.** To numerically test our theoretical predictions, the network is simulated utilizing the Neural Simulation Tool program (NEST) [36] and the EDLIF neuronal model as well as the ED-STDP synaptic model are specified using NESTML [37], the domain-specific language tailored for the spiking neural network simulator NEST. Models and code are available on GitHub (https://github.com/Wiss/edsnn).

The simulated network has $n = 500$ neurons, and the excitatory-inhibitory ratio is $n_E : n_I = 4 : 1$, following biologically realistic excitatory-inhibitory ratio values [38].

Given the need for a mathematically tractable framework, the network architecture is defined with an all-to-all structural connectivity. In addition, following *in vitro* measured weights strength in neuronal cell assemblies [39], initial synaptic strength values for the simulated network are drawn from an exponential distribution (see Eq 28)

$$f(w; \beta) = \begin{cases} \frac{1}{\beta}e^{-w/\beta}, & \text{if } w \geq 0, \\ 0, & \text{otherwise,} \end{cases} \tag{28}$$

where $f(w; \beta)$ is the probability density function of the exponential distribution. This means that the distribution of the weights $w$ is described by an exponential distribution with scale parameter $\beta$. Therefore, the expected value of the weights is $\text{E}[w] = \beta$ and the variance $\text{Var}[w] = \beta$. For the simulation we used $\beta = 5$.

To emulate the incoming activity to each neuron from other brain areas, we inject into each neuron a constant current $I_{\text{stim}}$ drawn from a Gaussian distribution with mean $\mu = 166\,pA$ and standard deviation $\sigma = 15\,pA$. By assuming that all neurons share the same parameters values we study the homogeneous parameter scenario (neuronal parameters are in S1 Table).

## Results

### Energy-dependent plasticity and equilibrium

While studying static synapses and energy consumption in the neuron, closed-form expressions for available energy level in the cell were introduced. In Eq 16 the neuron's energy expenditure due to static synaptic activities are pondered by the normalized synaptic strength $\tilde{w}_k$, however strength between neurons change over time (*i.e.*, $w_k := w_k(t)$). Here we analytically explore how the available energy in the neuron jointly with the synaptic sensitivity to energy imbalances $\eta$ impact the synaptic plasticity between neurons. Specifically, we try to understand how metabolic constraints dictate synaptic plasticity equilibrium points, where synaptic potentiation and depression balance each other on average.

As we previously explained, our energy-dependent STDP rule uses postsynaptic available energy as a neuromodulator. Therefore, for low postsynaptic energy levels, potentiation is suppressed and the suppression depends on the sensitivity parameter $\eta$. For the following explanation let us assume equal synaptic time constants (*i.e.*, $\tau_+ = \tau_-$ in Eq 20). Intuitively, given Eq 20 and $A(t = 0) = A_H$ initial postsynaptic available energy, if peak potentiation is stronger than peak depression (*i.e.*, $\alpha \in [0, 1)$) then, for uniformly random pre- and postsynaptic spike times, in a short-time window simulation, weights tend to increase in average (see Fig 2A).

Higher weights induce more energy consumption in the postsynaptic neuron (because of the increased energy expenditure related to postsynaptic potentials and, also, because postsynaptic neurons will integrate more current, potentially increasing firing rate, which also contributes to energy consumption). Therefore, postsynaptic available energy decreases, suppressing potentiation. If potentiation is strongly suppressed then, for uniformly random pre- and postsynaptic spike times, in a short-time window simulation, depression will have a greater effect, generating a net depression effect on weights (see Fig 2B). This net depression force weights to decrease on average, thus decreasing the postsynaptic firing rate and also diminishing postsynaptic energy consumption. Consequently, postsynaptic available energy increases, favoring potentiation over depression, producing a potentiation depression loop (see Fig 2C). However, it is also possible that the postsynaptic available energy reaches an equilibrium level where potentiation and depression balance each other. Let us formally explain this intuition. If we assume random uniformly distributed inter-spike intervals $\Delta t = t_{post} - t_{pre}$:

$$\Delta t \sim U[-\Delta t', \Delta t']. \tag{29}$$

Given the postsynaptic available energy $A$, the net effect, also called drift, experienced by weights is the integral of $\Delta w$ from Eq 20 over all possible $\Delta t$:

$$\int_{-\Delta t'}^{\Delta t'} \Delta w\, dt = \int_{-\Delta t'}^{0} -\lambda \alpha e^{t/\tau_-}\, dt + \int_{0}^{\Delta t'} \lambda e^{-\eta(A_H - A)/A_H} e^{-t/\tau_+}\, dt \tag{30}$$

$$= \lambda \left[ -\alpha \tau_- \left(1 - e^{-\Delta t'/\tau_-}\right) - \tau_+ e^{-\eta(A_H - A)/A_H}\left(1 - e^{-\Delta t'/\tau_+}\right) \right] \tag{31}$$

If $\tau_+ = \tau_- = \tau_{syn}$, Eq 31 reads:

$$\int_{-\Delta t'}^{\Delta t'} \Delta w\, dt = \underbrace{\lambda \tau_{syn}\left(1 - e^{-\Delta t'/\tau_{syn}}\right)}_{\geq 0}\left[-\alpha + e^{-\eta(A_H - A)/A_H}\right]. \tag{32}$$

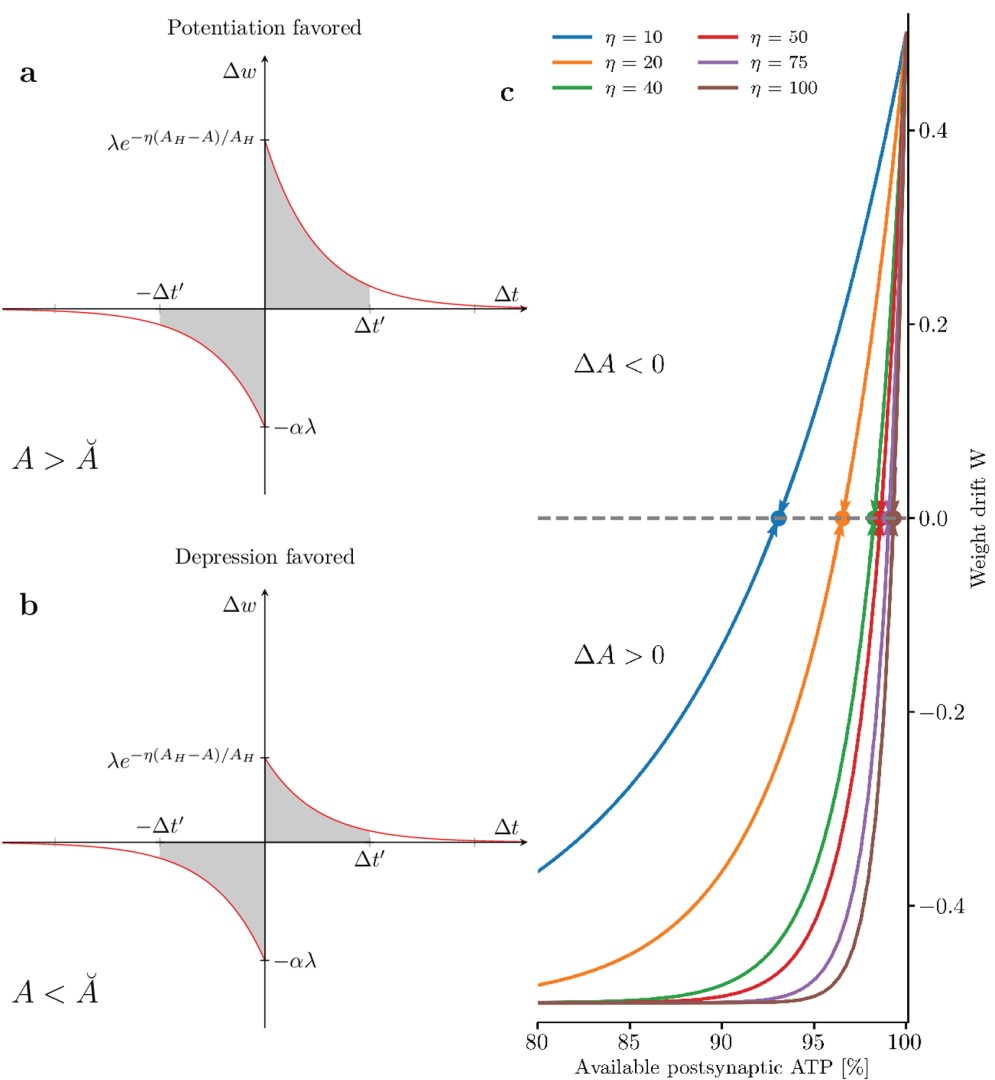

**Fig 2. Energy-dependent plasticity and postsynaptic available energy.** Potentiation suppression depends on postsynaptic energy level $A$. If $A > \breve{A}$ potentiation is favored **(a)**, while for $A < \breve{A}$ depression is favored **(b)**. **(c)** The weight drift Eq 32 is shown for varying levels of postsynaptic available energy $A$ and different synaptic energy imbalances sensitivity $\eta$. The points of intersection between the horizontal gray line and the colored curves indicate the zero drift and equilibrium energy level $\breve{A}$ for different $\eta$ values.

Consequently, if $e^{-\eta(A_H-A)/A_H} > \alpha$, then potentiation is favored (*i.e.*, $\int_{-\Delta t'}^{\Delta t'} \Delta w \, dt \geq 0$). Whereas if $e^{-\eta(A_H-A)/A_H} < \alpha$ depression is favored. Similarly, by imposing $e^{-\eta(A_H-A)/A_H} = \alpha$ we can find the update weight fixed point (*i.e.*, $\Delta w = 0$), which gives the energy level $\breve{A}$ value for which potentiation and depression mutually balance each other:

$$\breve{A} = A_H\left(1 + \frac{\ln(\alpha)}{\eta}\right). \tag{33}$$

Eq 33 predicts the available energy level on the postsynaptic neuron for which, on average, potentiation and depression are compensated.

Moreover, the equilibrium level for postsynaptic available energy $\breve{A}$ is inversely proportional to the sensitivity parameter $\eta$. As a consequence, if the plasticity rule is highly sensitive to energy imbalances, then Eq 33 guarantees that the postsynaptic energy level stay close to $A_H$ (see Fig 3).

As we will explain, this fixed point is crucial for the network dynamics, because for $A = \breve{A}$ the weights are static (on average), thus the average incoming current to the neuron should stay approximately constant if the average is calculated over a time period $T_w$ greater than membrane and synaptic time constant (*i.e.*, $T_w \gg \tau_m, \tau_{syn}$). Thus, we should expect a fixed point for the firing rate.

## Network's energy dynamic and fixed points under metabolic constraints

Built on the energy-dependent single-neuron model and the energy-dependent synaptic plasticity model, now we formalize, simulate, and investigate the dynamic and structure of E-I networks including energy constraints in both, the single-neuron model as well as the synaptic plasticity. The developed models allow investigation on how different sensitivities to energy imbalances at single neuron and synaptic level affect the network's dynamics and structure.

**Energy dynamic and fixed point analysis.**   When studying the network's dynamics, we are particularly interested in knowing if the network converges or oscillates toward a specific state. If the network evolves toward an specific state independently of the initial conditions, those states are attractors of the network. Therefore, to find fixed points in the network dynamics, we start by trying to find if there are energy attractors in the network dynamics, when all the neurons share the same parameters (the homogeneous case). Accordingly, we analyze stable fixed points for energy states of an arbitrary neuron in the network.

The total energy change $\Delta A(t) = A(t + T) - A(t)$ of any neuron in the network in a time interval $T$ is calculated by summing the contributions of energy production and consumption

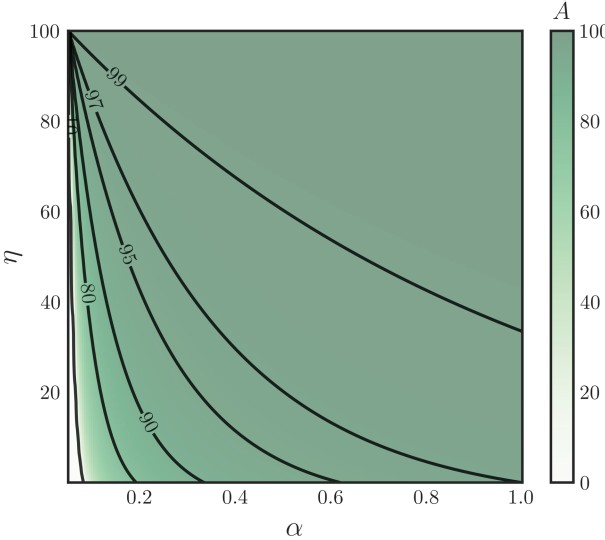

**Fig 3. Available postsynaptic energy equilibrium level $\breve{A}$.** Postsynaptic energy equilibrium level $\breve{A}$ as a function of the sensitivity parameter $\eta$ and depression scaling factor $\alpha$. $A_H = 100\%$.

occurring in the time interval $[t, t + T]$. From Eq 6 we obtain:

$$\Delta A(t) = \int_t^{t+T} A_s(t') - C(t') dt'. \qquad (34)$$

Previously, we formalized the synaptic energy consumption $C_{syn}$ (see Eq 11) as well as the neuron's own action potential energy consumption $C_{ap}$ (see Eq 13), for arbitrary $\varepsilon_{syn}$ and $\varepsilon_{ap}$ kernels. To simplify the mathematical tractability, now we will choose the delta kernel $\varepsilon(t - t_s) = \delta(t - t_s)$ for $\varepsilon_{syn}$ and $\varepsilon_{ap}$. Thus, Eq 34 reads:

$$\frac{1}{T}\Delta A(t) = \frac{1}{T}\int_t^{t+T} K(A_H - A(t'))dt' - E_{ap}\frac{1}{T}\int_t^{t+T}\sum_{\text{spike } s}\delta(t' - t_s)dt' -$$

$$E_{syn}\frac{1}{T}\int_t^{t+T}\sum_{\text{synapse } k}\sum_{\text{spike } s}\tilde{w}^k(t')\delta(t' - t_s)dt'. \qquad (35)$$

Because weights change slowly (*i.e.*, $\lambda \ll 1$ in Eq 20), for small $T$ we can assume they are constant in the $[t, t + T]$ interval (*i.e.*, $\tilde{w}^k(t) = \tilde{w}_t^k$). Formally, we are assuming that time scales of learning and neuronal spike dynamics can be separated [40]. In addition, because of the linearity of the integral operator:

$$\frac{1}{T}\Delta A(t) = K(A_H - \overline{A}_t) - E_{ap}\frac{1}{T}\sum_{\text{spike } s}\delta(t - t_s) - E_{syn}\frac{1}{T}\sum_{\text{synapse } k}\sum_{\text{spike } s}\tilde{w}_t^k\delta(t - t_s)$$

$$= K(A_H - \overline{A}_t) - E_{ap}\overline{\boldsymbol{\nu}}_t - E_{syn}\sum_{\text{synapse } k}\tilde{w}_t^k\overline{\boldsymbol{\nu}}_t^k, \qquad (36)$$

where $\overline{A}_t = T^{-1}\int_t^{t+T} A(t')dt'$ is the mean energy level of the neuron in the $[t, t + T]$ interval, and in the second and third terms we obtain the neuron self and connected neighbor's firing rates, respectively. Now, if we impose $\Delta A = 0$ in Eq 36, it is possible to find neuron's firing rates, incoming weight, and energy expenditure values that satisfy the energy fixed point constraints:

$$\overline{A}_t = A_H - \frac{1}{K}\left[ E_{ap}\overline{\boldsymbol{\nu}}_t + E_{syn}\sum_{\text{synapse } k}\tilde{w}_t^k\overline{\boldsymbol{\nu}}_t^k \right], \qquad (37)$$

which in matrix form reads:

$$\overline{\boldsymbol{A}}_t = \boldsymbol{A}_H - \frac{1}{K}\left[ E_{ap}\mathbb{I} + E_{syn}\tilde{\mathbf{w}}_t \right]\overline{\boldsymbol{\nu}}_t, \qquad (38)$$

where $\overline{\boldsymbol{A}}_t = [\overline{\boldsymbol{A}}_t^{E\top}, \overline{\boldsymbol{A}}_t^{I\top}]^\top$. Eq 37 holds if the neuron is in an energy fixed point and, to achieve the energy fixed point, the excitatory-excitatory connections to the neuron under study also need to achieve a fixed point (the other connections are static). Otherwise, the synaptic energy consumption ($C_{syn}$) will stay varying, impeding to achieve an energy fixed point in the postsynaptic excitatory neuron. Please realize that, if the excitatory-excitatory connections achieve a fixed point as well as the presynaptic neuron's firing rate, then the firing rate of the postsynaptic neuron also achieves a fixed point (*i.e.*, $\Delta\overline{\boldsymbol{\nu}}_t = 0$).

Excitatory-excitatory connections stabilize at a fixed point when the postsynaptic excitatory neuron's energy level matches $\check{A}$. However, for silent neurons incoming weights remain

unchanged, leading to a constant energy level in the neuron ($\Delta A = 0$), regardless of whether $\overline{A}_t \not\approx \breve{A}$. Therefore, silent neurons (neurons with firing rate $\overline{\nu}_t$ equal to zero) conform to Eq 37. If there is no rise in their incoming current, they will stay at the fixed point where $\Delta A = \Delta w = \Delta \nu = 0$, even without needing to meet the $\overline{A}_t \approx \breve{A}$ condition. Thus, for non-silent neurons in the fixed point, we can approximate $\overline{A}_t \approx \breve{A}$, obtaining:

$$\breve{A} \approx A_H - \frac{1}{K}\left[ E_{ap}\overline{\boldsymbol{\nu}}_t + E_{syn} \sum_{\text{synapse } k} \tilde{w}_t^k \overline{\boldsymbol{\nu}}_t^k \right]$$

$$A_H\left(1 + \frac{\ln(\alpha)}{\eta}\right) \approx A_H - \frac{1}{K}\left[ E_{ap}\overline{\boldsymbol{\nu}}_t + E_{syn} \sum_{\text{synapse } k} \tilde{w}_t^k \overline{\boldsymbol{\nu}}_t^k \right]. \tag{39}$$

By rearranging Eq 39, we arrive to the following relationship constraining weights and firing rates of each neuron in the network:

$$-\frac{A_H K \ln(\alpha)}{\eta} \approx E_{ap}\overline{\boldsymbol{\nu}}_t + E_{syn} \sum_{\text{synapse } k} \tilde{w}_t^k \overline{\boldsymbol{\nu}}_t^k. \tag{40}$$

If all the neurons in the network converge towards a fixed point, they must satisfy Eq 37 and, in particular, non-silent excitatory neurons must satisfy Eq 40. In consequence, in this state, all the excitatory-excitatory connections in the network remain constant (*i.e.,* $\Delta\tilde{w}_t = 0$) in average as well as the firing rates (*i.e.,* $\Delta\overline{\boldsymbol{\nu}}_t = 0$). As a result, Eq 40 allow us to predict the energy consumption of each non-silent excitatory neuron in the network after converging towards a fixed point.

The energy consumption induced by action potentials can be expressed in terms of the energy consumption induced by synapses *i.e.* $E_{ap}\overline{\boldsymbol{\nu}}_t \approx mE_{syn}\sum_k \tilde{w}_t^k \overline{\boldsymbol{\nu}}_t^k$ (experimental findings indicate that $m$ is approximately 1/3 [41]). Thus, Eq 40 can be rewritten as:

$$-\frac{A_H K \ln(\alpha)}{(m+1)\eta E_{syn}} \approx \sum_{\text{synapse } k} \tilde{w}_t^k \overline{\boldsymbol{\nu}}_t^k$$

$$= \sum_{\text{synapse } k} \tilde{w}_t^{EE,k} \overline{\boldsymbol{\nu}}_t^{E,k} + \sum_{\text{synapse } k} \tilde{w}_t^{EI,k} \overline{\boldsymbol{\nu}}_t^{I,k}. \tag{41}$$

Consequently,

$$\underbrace{-\frac{A_H K \ln(\alpha)}{(m+1)\eta E_{syn}} - \sum_{\text{synapse } k} \tilde{w}_t^{EI,k} \overline{\boldsymbol{\nu}}_t^{I,k}}_{\Lambda} \approx \sum_{\text{synapse } k} \tilde{w}_t^{EE,k} \overline{\boldsymbol{\nu}}_t^{E,k}. \tag{42}$$

Eq 37 reveals a metabolic constraint over incoming synapse strengths and their respective presynaptic neuronal firing rates, affecting non-silent excitatory neurons in the energy fixed point. In particular, Eq 40 shows that, to achieve a metabolic fixed point in the excitatory postsynaptic neurons, there is a trade-off between synaptic strength and the corresponding presynaptic neuronal firing rate. Interestingly, the metabolic constraint in Eq 40 dictates an inverse relationship between weights and firing rates, which is a completely new constraint in the system, given that previous models generally do not account for metabolic activity. Moreover, the constraint mentioned above enables the emergence of an attractor state to the

network, as a consequence of the interception between the metabolic and the physically plausible states of the network is given by the previously introduced metabolic (see Eq 40) and weight-frequency relations (see Eq 25).

On the one hand, Eq 25 gives a weight-rate relationship where increasing excitatory-excitatory weights $\mathbf{w^{EE}}$ generates higher excitatory firing rates $\nu^E$, due to higher mean incoming synaptic currents to postsynaptic neurons. On the other hand, if either excitatory-excitatory weights or excitatory firing rates increase, the postsynaptic energy levels drop (see Eq 38). In addition, at the metabolic fixed point and under previously described assumptions, non-silent excitatory neurons fulfill Eq 42. As a consequence, if excitatory firing rates (excitatory-excitatory weight) increase, then excitatory-excitatory weights (excitatory firing rates) must decrease. Thus, Eq 42 imposes an inverse relationship between excitatory-excitatory weights and excitatory firing rates, contrary to Eq 25, where the relationship between the two variables is direct. Fig 4 shows a conceptualization of the previous reasoning.

With an analytical understanding of how metabolic constraints affect the structure and dynamics of the network, let us now explore the impact of metabolic constraints on the network through numerical simulations.

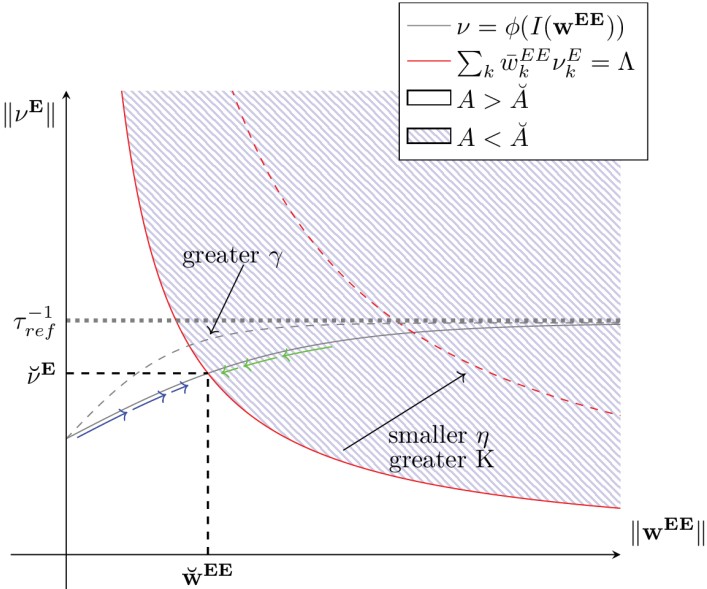

**Fig 4. Energy-dependent E-I network and constraints intersection.** The gray curve describes excitatory firing rate magnitude as a function of excitatory-excitatory weights magnitude (see Eq 25), whereas the red curve conceptualizes the inverse relation between excitatory firing rates and excitatory-excitatory weights magnitudes given by metabolic constraints (see Eq 42) affecting non-silent excitatory neurons in the energy fixed point. On the blue region, the postsynaptic energy level drops below the energy level fixed point $\breve{A}$. Thus, in the blue region, $\mathbf{w^{EE}}$ experience a net depression drift (see Eq 31), whereas in the white region, the postsynaptic energy level is above $\breve{A}$. Consequently, in the white region, there is a net potentiation drift. Therefore, if the network state is in the blue region, energy-dependent plasticity rules push the network state toward the intersection between the two curves (following the green arrows). Likewise, if the network is in the white region, energy-dependent plasticity pushes the network state toward the intersection point between the two curves (following the blue arrows). Finally, in the intersection between the two curves (the system's metabolic fixed point) the postsynaptic energy level is $\breve{A}$ and, as a consequence, $\Delta \mathbf{w^{EE}} = 0$ on average. For a detailed fixed point stability analysis, please refer to S1 Text.

**Network activity and structure without energy constraints.**  Before analyzing the network's structure and dynamics under energy constraints on simulations, we first simulate the network without energy constraints (*i.e.,* $\gamma = \eta = 0$). Thus, the current-rate relation from Eq 25 as well as the available energy in the neuron dictated by Eq 38 holds, but Eq 40 does not hold, because Eq 40 holds for the excitatory population only if energy constraints are affecting $\mathbf{w^{EE}}$ plasticity (*i.e.,* $\eta > 0$).

Simulating the network without metabolic constraints is useful as a base case for comparisons before including metabolic constraints (all the simulations presented in the Results section have the same initial conditions). Fig 5 shows the network's dynamics and structure when no metabolic constraints are included, with panels **(b)** and **(c)** included for direct comparison to simulation with metabolic constraints. As expected, firing rates increase until saturation due to favored synaptic potentiation ($\alpha = 0.5$). In this regard, the refractory period for each neuron in the network is $\tau_{ref} = 8\ ms$. Thus, the maximum theoretical firing rate for an infinite stimulation current is $\tau_{ref}^{-1} kHz \Rightarrow \nu_{max} = 125Hz$ (see Eq 21). Therefore, the network's firing rate is close to the saturation state. Moreover, the mean energy level of the excitatory population (Fig 5) decreases until stabilizing at $A(t) \approx 75\%$. The energy stabilization point occurs at the firing rate and excitatory-excitatory strength saturation.

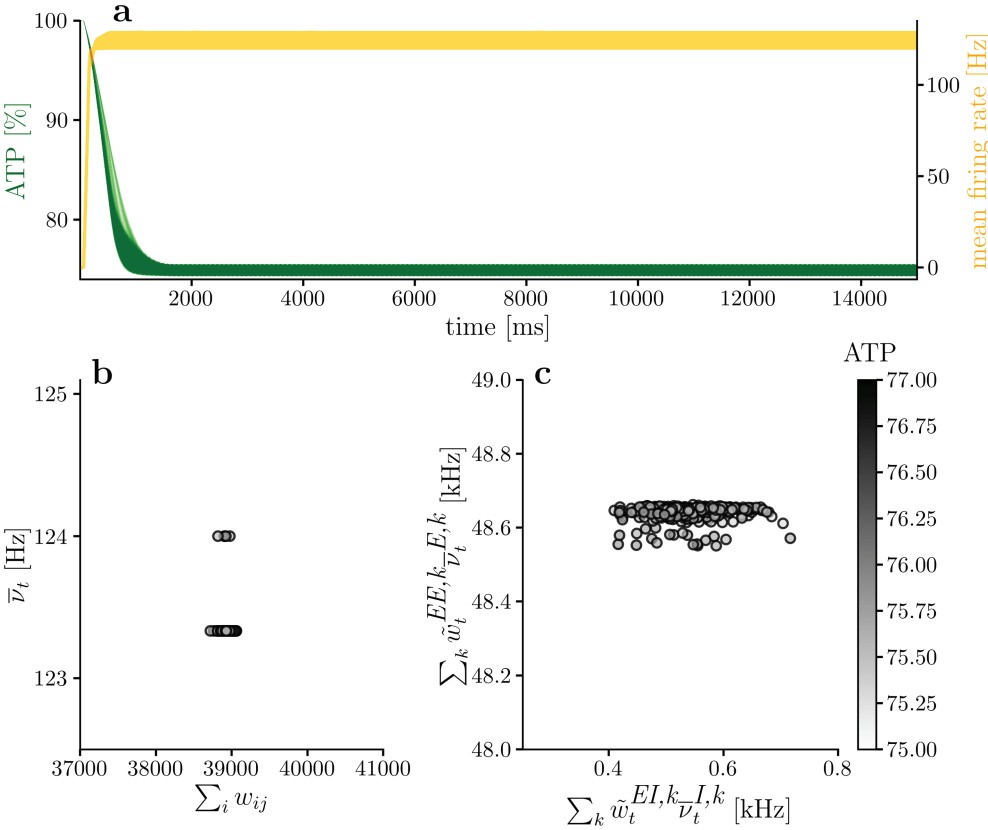

**Fig 5. Network dynamic and structure without energy constraints.** Network simulation when there are no metabolic constraints in the network. Mean available energy in the excitatory population keeps dropping until neuronal firing rates are saturated. **(a)** shows mean energy level and mean firing rate for the excitatory population, while **(b)** shows final incoming synaptic strength per neuron ($\sum_i w_{j,i}$) and mean firing rate per neuron. The mean firing rates in **(b)** and **(c)** are calculated considering the last 10% of the simulation.

The previous observation is supported by realizing that if firing rates and excitatory-excitatory weights are saturated, the energy consumption in each neuron can be approximated by:

$$C \approx E_{syn} \sum_k \tilde{w}_k \overline{\boldsymbol{\nu}}_k$$
$$\approx E_{syn} \times n_E \times \tilde{w}_{max} \times \nu_{max}$$
$$= 0.5 \times 400 \times 1 \times 0.125 = 25\%/ms,$$

in agreement with what is observed in Fig 5A. However, in principle, without energy constraints, each neuron's energy level can decrease until achieving $A = 0$. This is the case if we, for instance, sufficiently increase the number of neurons in the network, or increase the energy consumption related to postsynaptic potentials $E_{syn}$. Eq 37 describes which variables affect energy consumption and the available energy in the neuron.

However, we do not analyze the $A \to 0$ limit in this work, as it is outside the scope of our study.

**Exploring synaptic energy imbalances sensitivity $\eta$.** From our previous theoretical developments, for higher synaptic energy imbalance sensitivities $\eta$ we expect higher energy level equilibrium values $\breve{A}$ (Eq 33). To achieve $\breve{A}$ closer to $A_H$, energy consumption needs to decrease, and postsynaptic energy consumption decreases if the presynaptic rate decreases, or if incoming synaptic strengths decrease (Eq 41). Therefore, as we increase $\eta$, we expect to observe both lower firing rates and weaker excitatory-to-excitatory synaptic strengths. This trend is conceptually illustrated in Fig 4 and is numerically validated by the data presented in the first row of S3 Fig. In addition, to satisfy Eq 41 an inverse relation between $\sum_k \tilde{w}_t^{EI,k} \overline{\boldsymbol{\nu}}_t^{I,k}$ and $\sum_k \tilde{w}_t^{EE,k} \overline{\boldsymbol{\nu}}_t^{E,k}$ should be observable in the simulations.

Fig 6 shows the available energy per neuron and mean neuronal firing rate through the simulation, for the excitatory population with different synaptic energy imbalance sensitivity ($\eta$). Consistently with Eq 33, while synaptic energy imbalance sensitivity $\eta$ increases, the energy level equilibrium point increases and stays close to our analytical prediction (dashed line). Also, while the energy fixed point $\breve{A}$ increases, the mean neuronal firing rate decreases, which aligns with the theoretical description represented in Fig 4.

Fig 6C shows neuronal firing rate as a function of incoming synaptic strength, for different synaptic sensitivities $\eta$ values. In agreement with Eq 25, as the incoming synaptic strengths increases, the firing rate increases. Also, when $\eta$ increases, neuronal rates and excitatory-excitatory weights decrease, as can be observed from Fig 6C with $\eta = 30$ (blue) having higher firing rates and stronger excitatory-excitatory synapses than the ones observed when $\eta = 50$ (purple). The previous observation is also valid when contrasting Fig 6C for $\eta = 50$ against $\eta = 100$ (red). Thus, when synaptic sensitivity $\eta$ increases, excitatory firing rates tend to decrease.

Regarding the inverse relation between $\sum_k \tilde{w}_t^{EI,k} \overline{\boldsymbol{\nu}}_t^{I,k}$ and $\sum_k \tilde{w}_t^{EE,k} \overline{\boldsymbol{\nu}}_t^{E,k}$ dictated by Eq 41, Fig 6D shows the relationship between these two quantities for simulations with increasing $\eta$. For a low synaptic sensitivity to energy imbalance ($\eta = 30$ (blue)), there seems to be an inverse relation, although not very clear. However, if synaptic sensitivity $\eta$ increases, the inverse relationship between the two variables become more evident, as shown in the $\eta = 50$ case (purple). Interestingly, when the synapses are highly sensitive to energy imbalance (the $\eta = 100$ case (red) in Fig 6C and Fig 6D), it is not possible for all the neurons to achieve the metabolic fixed point and some neurons remain silent (neurons with firing rate equal to zero in Fig 6C).

It is not surprising that silent neurons have higher inhibitory weights. This happens because just before the separation of the network into two subpopulations ($t \approx 7000ms$), there

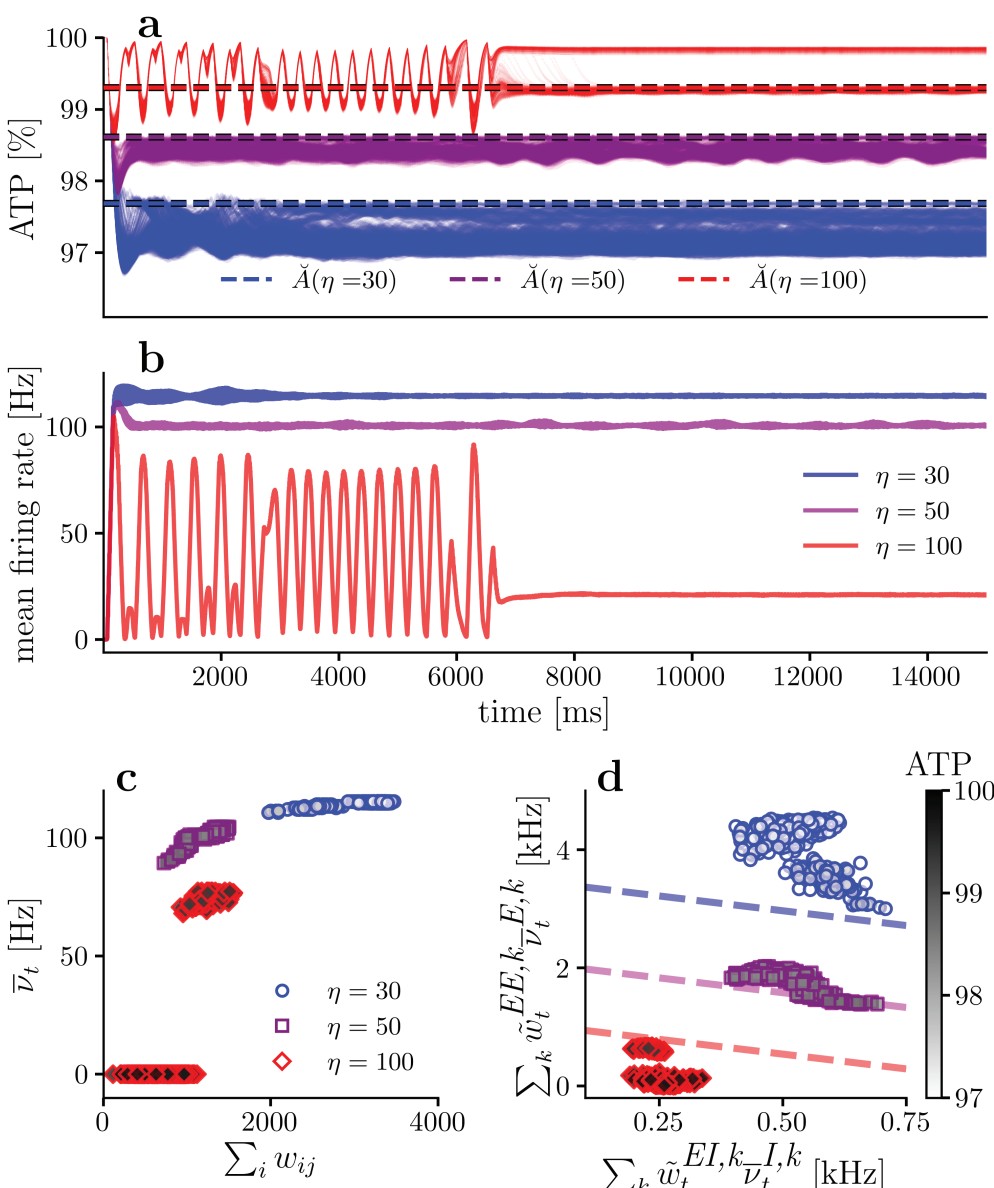

**Fig 6. Available energy and firing rate for different synaptic sensitivities to energy imbalances. (a)** Available energy per neuron for different synaptic sensitivities $\eta$. **(b)** Mean firing rate for the excitatory population for different synaptic sensitivities $\eta$. **(c)** Mean firing rate versus incoming synaptic strength per neuron. **(d)** Mean energy consumption per neuron due to presynaptic excitatory and inhibitory neurons, and the dashed lines represent the theoretical relationship described in Eq 41. In all simulation there is no neuronal sensitivity to energy imbalances (*i.e.,* $\gamma = 0$). In **(c)** and **(d)**, only the last 10% of the simulation is considered.

is a slight decrease in available energy coexisting with an increase in the firing rate, followed by an increase in available energy joined with a reduction in the firing rate. This last increment in available energy must be accompanied by a general decrease in excitatory-excitatory weights ($A < \check{A}$ implies a net negative drift in excitatory-excitatory weights, Eq 32). If there is a general decrement in excitatory-excitatory weights, then the first neurons to become silent are

those with higher inhibitory incoming weights. After entering the silent state, their excitatory-excitatory weights remain constant, so the only possibility for them to stop being silent is to receive higher incoming current, but this is not possible, because the silent neurons allow a decrease in the energy consumption in the non-silent neurons, enabling the non-silent neurons to satisfy the metabolic fixed point where $A \approx \breve{A}$. As a consequence, the silence of those neurons enables the network to converge toward a stable fixed point, with one subpopulation staying in the metabolic fixed point, while another subpopulation remains in the silent fixed point. In this scenario, one group of neurons still tries to satisfy the $\overline{A}_t \approx \breve{A}$ energy constraints, and their the energy equilibrium point is close to the one predicted by Eq 40, as shown in red in Fig 6A. The second group of silent neurons has a higher available energy, but their energy consumption is not zero. This is not the only case in which there must be silent neurons to achieve a global fixed point. For instance, if the neuronal population is large, then this phenomenon also occurs (although there is no need for extreme synaptic sensitivity to energy imbalances in this case), or if a metabolic impairment is simulated, as we will show later.

Under certain conditions, the separation of excitatory populations into two subpopulations can be interpreted as a specialization. Surprisingly, under the developed framework, silent neurons emerge as a consequence of local energy constraints in the network. If these silent neurons are not present, then it is not possible to achieve a global fixed point in the network.

**Including neuronal sensitivities to energy imbalances $\gamma$.** So far, the developed theoretical analysis of the energy-dependent network does not account explicitly for neuronal energy imbalance sensitivity $\gamma$. However, we know that neuronal sensitivity to energy imbalances $\gamma$ affects neuronal firing rate. Thus, we can include the neuronal sensitivity parameter in the current-rate mapping $\phi(\cdot) := \phi_\gamma(\cdot)$. In particular, if the energy level in a neuron is below the homeostatic energy level (*i.e.*, $A \leq A_H$), for the same incoming input current, the neuron's firing rate is higher for higher $\gamma$, as demonstrated by Eq 5 and shown in Fig 1. Consequently, if two networks ($A$ and $B$) are equal (in particular, have the same initial excitatory-excitatory weights), but the neurons in network $A$ have higher $\gamma$ sensitivity than neurons in network $B$, then network $A$ must have higher or equal excitatory firing rates than network $B$. As a consequence, to achieve an energy equilibrium point, network $A$ needs to decrease excitatory-excitatory synaptic strengths. This idea is conceptualized in Fig 4 with the gray dashed line representing the current-rate mapping when $\gamma$ is increased. Following the previous explanation, for higher $\gamma$ we expect the network to have weaker excitatory-excitatory synaptic strengths as well as higher firing rates.

To numerically explore the effect of modifying $\gamma$, we maintain synaptic sensitivity $\eta = 50$ fixed and vary the neuronal sensitivity to energy imbalances $\gamma$. Because the synaptic sensitivity $\eta$ is constant for all cases in Fig 7 and the energy level fixed point is independent of the neuronal sensitivity parameter $\gamma$, the theory predicts that networks with different neuronal sensitivity $\gamma$ (and all other parameters being equal) should converge towards the same energy level. The previous prediction is confirmed by observing Fig 7, where for different $\gamma$ values, the energy fixed points are almost the same. However, as already anticipated by the theory, in the metabolic fixed point, higher $\gamma$ values are associated with higher mean excitatory firing rates $\overline{\nu}_t^E$. This prediction is confirmed by observing how increasing $\gamma$ shifts the dots to the upper-left corner in Fig 7, thus numerically demonstrating that the energy sensitivity parameter $\gamma$ affects the population's firing rate, but does not change the energy equilibrium point $\breve{A}$.

From Fig 7D, it is clear that the relationship defined by Eq 41 is satisfied, thus respecting the inverse relationship between $\sum_k \tilde{w}_t^{EI,k} \overline{\nu}_t^{I,k}$ and $\sum_k \tilde{w}_t^{EE,k} \overline{\nu}_t^{E,k}$.

Regarding excitatory-excitatory synaptic strength, simulations show that excitatory-excitatory weight slightly decreases as neuronal sensitivity $\gamma$ increases. This observation is supported by realizing that when $\gamma$ increases, the energy fixed point does not change, but the

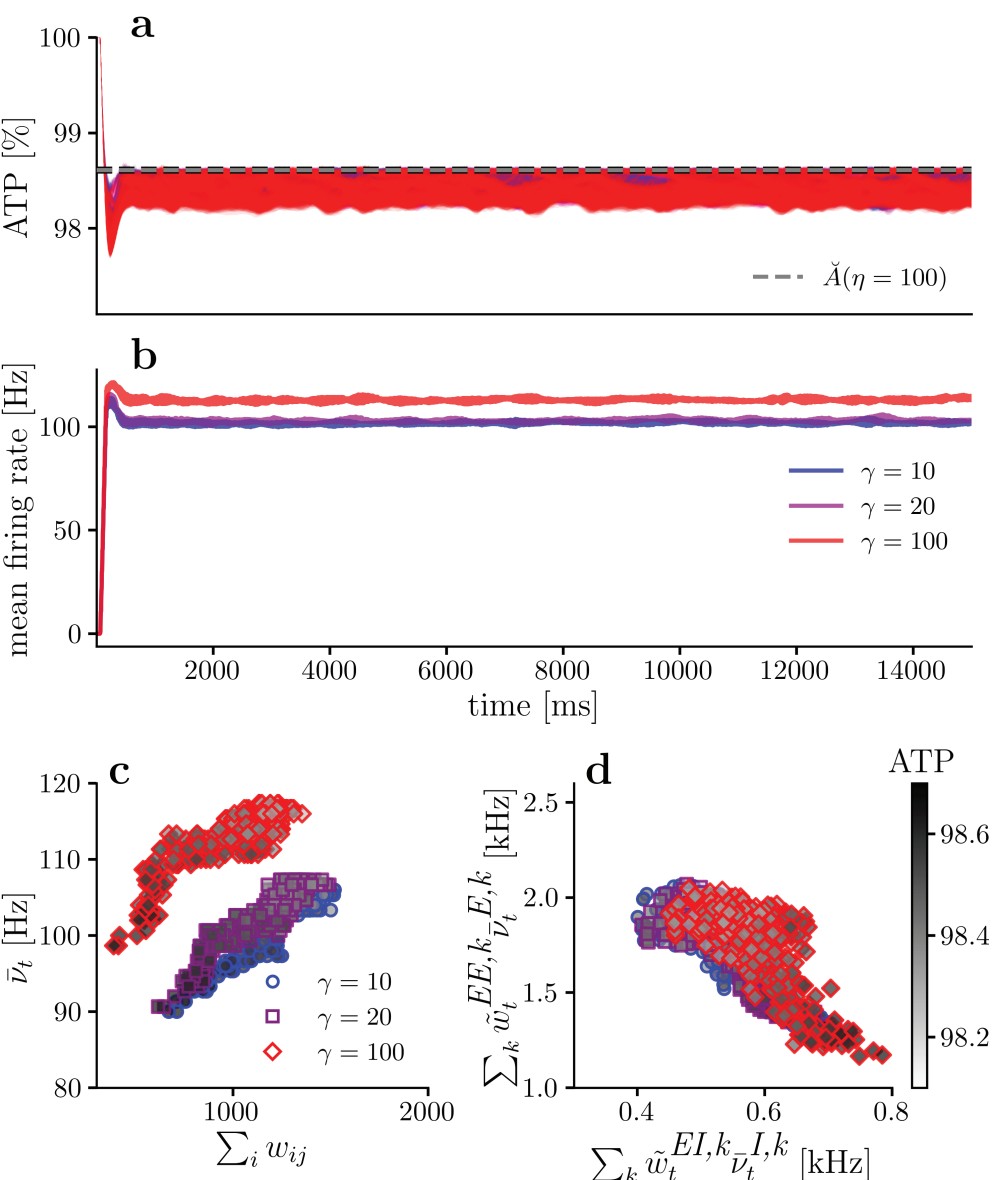

**Fig 7. Available energy and firing rate for different neuronal sensitivities to energy imbalances. (a)** Available energy per neuron for different synaptic sensitivities $\gamma$. **(b)** Mean firing rate for the excitatory population for different synaptic sensitivities $\gamma$. **(c)** Mean firing rate versus incoming synaptic strength per neuron. **(d)** Mean energy consumption per neuron due to presynaptic excitatory and inhibitory neurons, and the dashed lines represent the theoretical relationship described in Eq 41. In all simulation there is a constant synaptic sensitivity to energy imbalances ($\eta$ = 50). In **(c)** and **(d)**, only the last 10% of the simulation is considered.

firing rates increases (Fig 7A and 7B). This is only possible because the excitatory-excitatory weights decrease, compensating for the increased firing rates and thus allowing to keep the same energy consumption. Thus, the theoretical predictions regarding the effect of increasing $\gamma$ while keeping $\eta$ fixed is proved and in well agreement with numerical experiments.

**Impaired metabolic production.** Regarding biology, our theoretical and simulation framework allows us to study what happens when impaired metabolic production affects neurons. This question is relevant because evidence suggests that metabolic impairments

are a common cause for various neurodegenerative diseases [5–10]. Therefore, improving the understanding of how metabolic impairments affect network dynamics and structure is potentially valuable for clinical purposes.

From our theoretical developments, if the ATP rate production controlled by parameter $K$ decreases, we expect:

1. Given that the energy level equilibrium point $\breve{A}$ guaranteeing $\Delta w^{EE} = 0$ on average is independent of $K$ (see Eq 33), if there is a mean energy level fixed point for non-silent neurons, it should stay constant as $K$ varies.

2. By analyzing Eq 41, for non-silent neurons, if $K$ decrease, $\sum_k \tilde{w}_t^{EI,k} \overline{\nu}_t^{I,k}$ or $\sum_k \tilde{w}_t^{EE,k} \overline{\nu}_t^{E,k}$ needs to decrease in order to satisfy the aforementioned metabolic relation. The only plastic synapses in the network are $\mathbf{w^{EE}}$. Therefore, if the metabolic production is impaired, we expect lower firing rates as well as weaker excitatory-excitatory synaptic strengths.

3. In neurons with impaired metabolism (lower energy production rate $K$), slower responses to energy consumption create a delay between production and consumption, leading to oscillatory behavior in energy levels. When the initial available energy starts at the homeostatic level ($A(t = 0) = A_H$), but the energy fixed point is below $A_H$ ($\breve{A} < A_H$), energy dynamics undergo a cyclic process: excitatory-excitatory weights and firing rates initially rise, increasing energy consumption and causing available energy to drop. As energy falls below the fixed point, weights and firing rates decrease until energy production surpasses consumption, raising available energy above the fixed point. This feedback loop continues until energy production and consumption align, stabilizing energy levels near the equilibrium. When neurons' energy production is more acutely impaired, the mismatch between energy production and consumption increases, resulting in more pronounced oscillations in the system. Our analysis further shows that for sufficiently small $K$, the eigenvalues of the network around the fixed point become complex conjugates, generating oscillatory dynamics (see S2 Fig).

For the metabolically impaired simulations we vary $K$, but keep the synaptic sensitivity to energy imbalances $\eta = 50$ and the neuronal sensitivity $\gamma = 20$ (purple traces in Fig 7 show the *healthy* base case for comparison, where there is no metabolic impairment in energy production). In Fig 8 it is possible to observe that for a metabolic production rate of $K = 0.7$, the energy fixed point is the same as the one obtained in the healthy case (Fig 7 (purple)), confirming that if there is an energy fixed point for non-silent neurons, it is independent of the ATP rate production $K$, in agreement with our first theoretical prediction.

Moreover, comparing Fig 7D ($\gamma = 20$) with Fig 8D ($K = 0.7$), both simulations satisfy the energy relationship defined by Eq 41 (dashed lines). However, in Fig 7D ($K = 1$) the mean excitatory energy consumption ($\sum_k \tilde{w}_t^{EE,k} \overline{\nu}_t^{E,k}$) is much higher than in Fig 8D ($K = 0.7$), aligning with the second theory prediction. Additionally, contrasting Fig 7B with Fig 8B, we observe a decrease in mean firing rate for the excitatory population when ATP production rate $K$ decreases, also supporting the second theory prediction. The decrease in excitatory population firing rate with decreasing $K$ is also evident when comparing Fig 7C ($\gamma = 20$, purple. Firing rates ranging from approximately 90 to 110 $Hz$) to Fig 8C ($K = 0.7$, blue. Firing rates ranging from approximately 50 to 90 $Hz$). Therefore, if energy production is impaired in any neuron, then the energy consumption of that neuron needs to decrease to achieve a fixed point.

Hence, if we keep decreasing $K$ (more severe production impairment), energy consumption needs to drop to be consistent with the theory. When $K = 0.5$ (purple in Fig 8), the mean

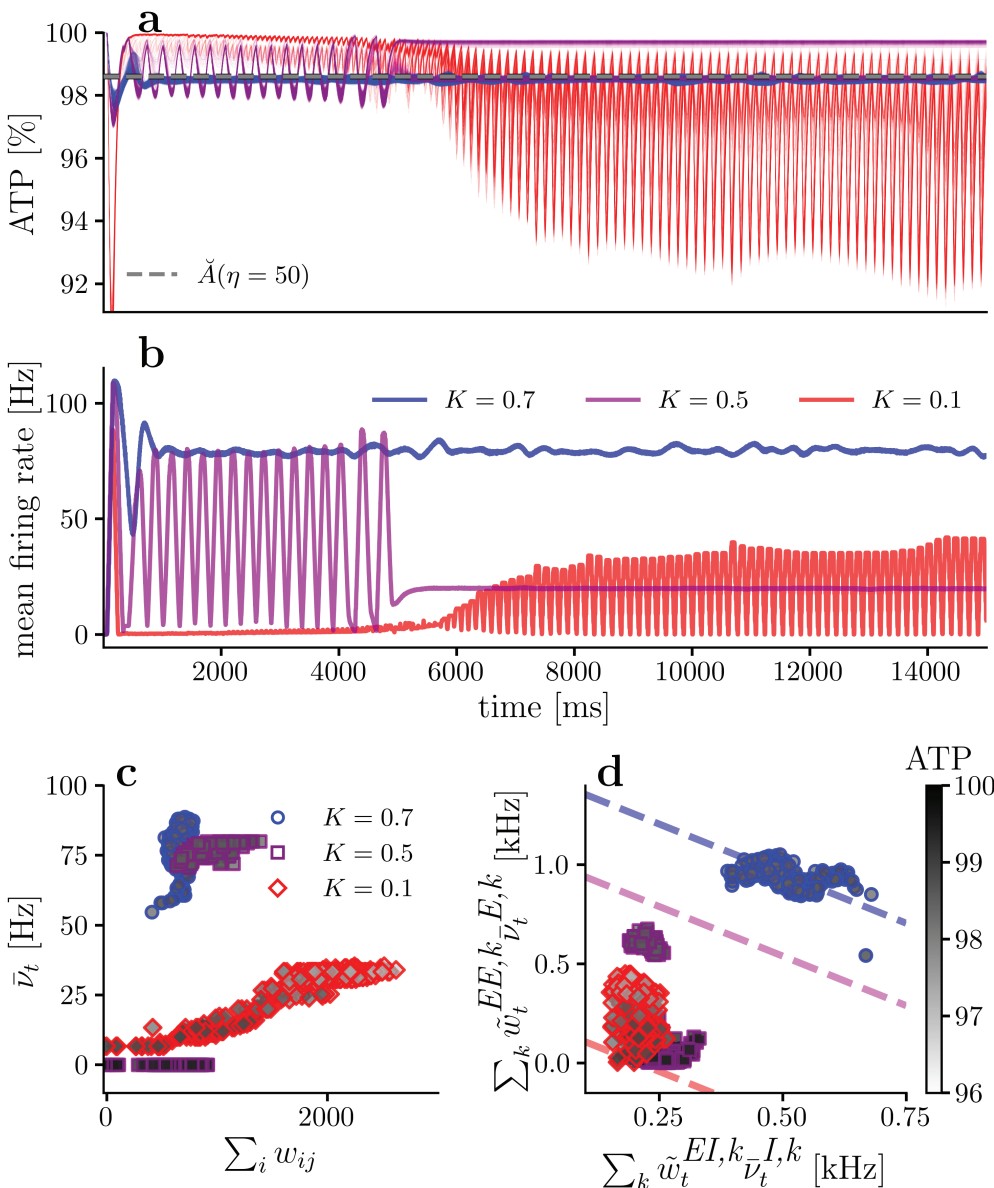

**Fig 8. Available energy and firing rate for different ATP production impairment.** (a) Available energy per neuron for different ATP production rates $K$. (b) Mean firing rate for the excitatory population for different ATP production rates $K$. (c) Mean firing rate versus incoming synaptic strength per neuron. (d) Mean energy consumption per neuron due to presynaptic excitatory and inhibitory neurons, and the dashed lines represent the theoretical relationship described in Eq 41. In all simulation there is a constant neuronal and synaptic sensitivity to energy imbalances ($\gamma = 20, \eta = 50$). In (c) and (d), only the last 10% of the simulation is considered.

firing rate and excitatory-excitatory weights decrease compared to the $K = 0.7$ case. However, the network can still converge to a global fixed point if a subpopulation of neurons becomes silent. This behavior aligns with the energy constraints of the network. As the ATP production parameter $K$ decreases, energy consumption must also decrease. To achieve this, excitatory-excitatory weights and mean excitatory firing rates decrease, resulting in a subpopulation of silent neurons to reduce energy consumption.

This phenomenon is similar to what happens when there is high synaptic sensitivity to energy imbalances ($\eta$ = 100 in Fig 7), but the cause is different. With high synaptic energy imbalance sensitivity ($\eta$), the energy level fixed point $\breve{A}$ is close to the homeostatic energy level $A_H$. This requires a reduction in energy consumption to reach a global fixed point.

In the case of metabolic impairment, the energy level fixed point $\breve{A}$ is not necessarily close to $A_H$. Here, energy consumption needs to drop because impaired energy production cannot meet high energy demands. This balance between energy consumption and production is necessary for the network to reach a fixed point. Therefore, the constraint in the metabolic impairment case is imposed by the neurons' energy production capability, not by synaptic energy sensitivity. Despite the different causes, the outcome is similar: silent neurons in both scenarios are those with higher inhibitory synapses.

This is because, before the network divides into two subpopulations, an increase in available energy is followed by a decrease in excitatory-excitatory weights. The first neurons to become silent are those with higher inhibitory weights as they receive less incoming excitatory current.

If we continue aggravating the metabolic production impairment and the network is able to achieve a fixed point, theoretically, we expect even weaker excitatory-excitatory synaptic strengths as well as lower firing rates in the equilibrium state. Accordingly, we simulate the network with $K$ = 0.1, emulating the case where neurons can produce energy at a 10% rate with respect to the healthy case (*i.e., K* = 1). Fig 8A and 8B show that the energy production is so impaired that the network is not able to achieve an equilibrium point and stay oscillating, exploring excitatory-excitatory weights strengths which allow for achieving an equilibrium state. The problem appears to be that there are no excitatory-excitatory weights values that allow a firing rate and energy consumption which can be compensated by the impaired energy production. Specifically, an abrupt change in firing rate due to a small variation in excitatory-excitatory strengths seem to be part of the problem. Thus, small excitatory-excitatory weight variations generate energy consumption that cannot be compensated by on-demand energy production. This is coherent with the third prediction and our S1 Text (S2 Fig), highlighting an extreme case where energy production is so impaired that it is not possible to converge towards a homeostatic balance, where energy production and consumption match each other.

To understand this intuitively, it is useful to remember that neurons with impaired metabolism (small $K$) have slower responses to energy consumption, causing a higher delay between consumption and production. Also, excitatory-excitatory weight changes depend on the available energy in the neurons (see Eqs 32 and 2).

If we start the simulation with all neurons at the homeostatic energy level $A(t = 0) = A_H$, and the energy fixed point is lower than the initial energy ($\breve{A} < A_H$), then initially, weights will increase, and firing rates will rise due to the current-rate relationship. This leads to increased energy consumption, but because energy production is delayed, the available energy will tend to decrease ($\Delta A(t) < 0$).

As available energy continues to drop, eventually it will be lower than the energy fixed point ($A(t) < \breve{A}$). When this happens, excitatory-excitatory weights will decrease, reducing firing rates. This process will continue until energy consumption decreases enough for energy production to surpass it. When energy production surpasses consumption, the available energy in neurons starts to increase ($\Delta A(t) > 0$) until it matches the energy equilibrium point ($A(t) = \breve{A}$).

Because energy production is always active below the fixed point ($A_s(t) = K(A_H - A(t))$), this allows the available energy to rise above $\breve{A}$. When the available energy exceeds the equilibrium value, the cycle repeats: weights and firing rates increase, leading to higher

energy consumption than the neuron can handle, causing another drop in available energy ($\Delta A(t) < 0$).

This cyclical behavior stops when energy production and consumption match, maintaining available energy near the fixed point ($A(t) \approx \breve{A}$). Achieving this equilibrium is easier when on-demand energy production can compensate for consumption in neurons. For further exploration of the impact of $K$ on network stability, please refer to S1 Text and S2 Fig.

Accordingly to our understanding of how energy constraint affects the networks, one possible solution to solve the non-converging system due to a dramatic ATP production impairment ($K = 0.1$), is decreasing the synaptic sensitivity to energy imbalances $\eta$. In this manner, synapses experience softer strengths updates transitions when they are close to the analytical fixed point $\breve{A}$. Thus, diminishing the oscillations due to changes in the peak energy-dependent potentiation described in Eq 20. To test this hypothesis, we simulate the network (Fig 9) with the same initial conditions and parameters as the one shown in Fig 8 with $K = 0.1$, but decreasing the synaptic sensitivity to energy imbalances to $\eta = 10$. The proposed solution allows for alleviating the oscillations in the system. However, the equilibrium is achieved by silencing a subpopulation of excitatory neurons. The logic behind this phenomenon is the same as the one explained in the $\eta = 50, \gamma = 20, K = 0.5$ case

Fig 9 shows that non-converging networks due to a dramatic metabolic impairment can be alleviated by modifying other parameters in the network. However, there are other solutions to this problem that might alleviate the oscillations. For instance, a simple solution is to decrease the energy expenditure due to postsynaptic potentials $E_{syn}$, thus forcing a decrease in energy consumption. Another possible solution could be modifying the neuron's time-constant $\tau_m$. In particular, if $\tau_m$ decreases, each neuron has a *slower* dynamic. Thus, decreasing the slope of the current-rate relation. As a consequence, new weights and rates combinations allow to satisfy the required energy constraints, but with lower rates, thus with slower weights updates (weights update when pre and postsynaptic spikes are present. Thus, lower rates imply fewer spikes and, therefore, slower weight updates). In addition, if the slope of the current-rate mapping decreases, weights modifications should produce softer transitions in rates, thus helping to decrease the oscillations due to the sensitivity of the firing rates to weights modifications.

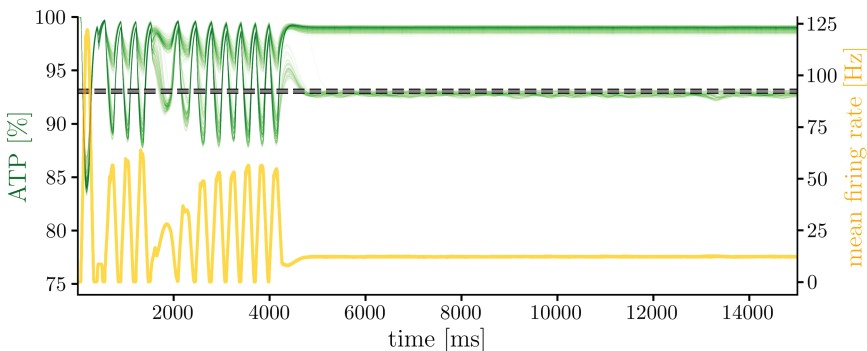

**Fig 9. Energy and firing rates for a dramatic metabolic impairment with low synaptic sensitivity to energy imbalances.** Simulations are obtained with $\eta = 10, \gamma = 20, K = 0.1$ parameters. Figure shows results for the excitatory population.

## Discussion

In this work, inspired by spike-timing-dependent plasticity models, we develop a phenomenological energy-dependent plasticity model that allows us to qualitatively replicate wet-biology experimental results, namely, the suppression of long-term potentiation when the available energy level on the postsynaptic neuron drops [29]. The model conditions resemble those observed in studies such as those from [29], and mainly *in vitro* experimental setups where metabolism is usually evaluated. Additionally, we derived a general mathematical expression for the available energy in every neuron in a network with arbitrary architecture.

### Energy-dependent plasticity model

The energy-dependent STDP model can be interpreted as a three-factor plasticity rule, where the postsynaptic available energy is modulating plasticity. The newly introduced model has a general structure that allows the implementation of both multiplicative and additive learning rules. However, we focus on the additive learning rules (*i.e.,* $\mu_\pm = 0$) and found a closed-form expression predicting the available energy level equilibrium point $\breve{A}$ on the postsynaptic neuron.

Our model focuses on the effect of postsynaptic available energy on the excitatory-excitatory synaptic potentiation, however, available energy possibly also affects depression. The mathematical formulation of ED-STDP in Eq 20 allows for extending the proposed model to account for energy dependencies on depression plasticity, and this is part of future work related to the energy homeostatic principle.

### Network dynamics under metabolic constraints

Based on the EDLIF model [20] and the ED-STDP plasticity rule introduced in this work, we mathematically formalized and simulated spiking neuronal networks under metabolic constraints. The work focuses on the emergence of dynamics, structure, and the study of attractors under energy constraints for homogeneous neuronal populations in E-I networks. In general terms, the developed theory allows us to predict behaviors observed in numerical experiments. Moreover, the introduced neuronal and synaptic energy constraints generate a new attractor in E-I networks due to the intersection of classic physical constraints (neuronal current-rate relations Eq 25) and the new constraints that emerge from the local energy constraint imposed on each neuron in the network. We mathematically describe this phenomenon and conceptualize it in Fig 4.

Regarding the network's fixed points, synaptic sensitivity to energy imbalances is the main parameter affecting the available energy fixed point for each neuron. In particular, the synaptic sensitivity ($\eta$) has a greater impact than the neuronal sensitivity ($\gamma$) on the network dynamics and fixed point. In fact, neuronal sensitivity $\gamma$ does not affect the energy equilibrium point (see Eq 33). However, because $\gamma$ modifies the neuron's activation function, if everything remains equal, different values of $\gamma$ will induce different weights and firing rates combinations to achieve the same energy equilibrium (Fig 4). This is shown in Fig 7, where different values of $\gamma$ do not change the energy equilibrium point (Fig 7A), while the specific firing rates and weight values change (Figs 7C–7D, and S3C–S3F) to achieve that equilibrium point. On the other hand, if we only change the synaptic sensitivity to energy imbalances ($\eta$), then the energy equilibrium point, as well as the weights and firing rates, changes (Figs 6 and S3A–S3C). This shows that the synaptic sensitivity ($\eta$) parameter has a greater impact on networks dynamics and fixed point compared to the neuronal sensitivity parameter ($\gamma$).

In our model, through energy-dependent plasticity, synaptic strengths change until non-silent neurons achieve the energy equilibrium point $\breve{A}$. When the weights change, the neurons' firing rates change accordingly. We mathematically describe the required constraints on the postsynaptic neuron between inhibitory and excitatory presynaptic energy consumptions to achieve the energy equilibrium point (Eq 41). Surprisingly, if synaptic sensitivity to energy imbalances ($\eta$) is too high, to achieve a global fixed point in the network, the excitatory population divides into two subpopulations, with one of them composed of silent neurons.

The occurrence of this phenomenon is consistent with biology [42], shedding light on why silencing excitatory neurons could be practical to diminish energy consumption in neuronal networks, thus allowing the network's convergence towards a fixed point. Moreover, this phenomenon can be interpreted as a specialization in the excitatory population. The appearance of a silent subpopulation of neurons is interesting because silent neurons are present in our brains, and it is not clear why we would have them. In fact, silent neurons have been referred to as "dark neurons" in analogy to the astrophysical observation that much of the matter in the universe is undetectable, or dark [42]. In our framework, silent neurons play an important role from a metabolic point of view. One hypothesis is that given local energy constraints in biological neural networks, the price to pay for having redundant neurons in the brain is to silence some of them to achieve a network energy equilibrium.

## Impact of metabolic impairment and neurodegenerative diseases

Regarding neurodegenerative diseases and given the biological evidence suggesting their relation with metabolic impairment, we study how neurons with impaired energy production affect the network dynamics and structure. Our theoretical developments predict that energy consumption needs to drop in cases of impaired energy production. Consequently, lower firing rates and excitatory-excitatory synapses are expected. These predictions are confirmed by numerical experiments (Fig 8). However, there are other important details in the metabolic impairment scenario. For instance, if the metabolic impairment is high enough, it is also possible that the excitatory population divides into two subpopulations, with one of them composed of silent neurons, although the source of this phenomenon (neuronal incapacity to produce enough energy on demand) is different from the one explained previously when too high synaptic energy sensitivity is present in the network (the energy fixed point $\breve{A}$ must be close to $A_H$, forcing low energy consumption). Even more dramatic energy production impairments may produce constant oscillations in the network, preventing the convergence toward an attractor ($K = 0.1$ case in Fig 8 and S2 Fig). We show how these oscillations can be alleviated by modifying other parameters of the simulations. Trying to alleviate the oscillations present in dramatic metabolic impairment scenarios is important because it could help in developing new treatments for neurodegenerative diseases. A detailed study of how to alleviate pathological behaviors due to metabolic impairments is out of the scope of this work. However, the proposed theory and the simulation framework could be valuable in deepening the knowledge about the relationship between neurodegenerative diseases and metabolic impairments at the neuronal, synaptic, and network levels.

It is critical to note that neurodegeneration is usually linked to mitochondrial dysfunction or oxidative stress [43]. Thus, neurodegeneration has been treated by increasing mitochondrial energy production or affecting activity through neurotransmission (e.g., pregabalin or levodopa to modulate activity). This study proposes an additional avenue for neurodegeneration treatment through synaptic weights. The challenge is to finely regulate energy expenditure and production rather than increase or decrease it in general terms, as currently done.

The rationale is that neurons have a mainly aerobic metabolism, with anaerobic activity delegated to astrocytes during increased synaptic activity [15]. Neurons are highly oxidative and some neuronal populations can be vulnerable to oxidative stress [44]. To our knowledge, they maintain a constant availability regardless of activity level [45] when applying stimulations that do not exceed physiological levels. Reducing synaptic weight is an important way for a neuron to regulate incoming activity from presynaptic neurons. Synaptic scaling experiments demonstrate this phenomenon by increasing activity through the addition of a GABAergic inhibitor (Bicuculline) to cultured cortical neurons for long periods (48 hours) [46,47]. Under this protocol, neurons transiently increase their mean firing rate but later return to a set point value by modulating synaptic weights [48]. This suggests that exploring potential drugs that boost synaptic weight update velocity may contribute to controlling neurodegenerative processes derived from oxidative stress. Also, our study stresses the need for a better understanding of the fined-tuned energy regulation systems in neurons for the development of new therapeutic avenues.

## Limitations and future directions

It is important to mention some of the limitations of this work. Firstly, we use a simple single-neuron model that neglects the neuron's morphology and the effect of some relevant ions, such as calcium kinetics, on the neuron's activity. Regarding the decision to use the EDLIF model, in our opinion, EDLIF is the simplest model including energy dependencies that allows for a detailed analytical exploration of the models' dynamics at the single-neuron and network level. However, there are other simple models that also include energy dependencies (eLIF and mAdEx [49], and Model 2 [50]). EDLIF, eLIF, and mAdEx share the same fundamental rationale, in the sense that a decrease in the available energy inhibits the Na/K pump function, thus leading to a sodium accumulation inside the cell. However, a different mechanism is used to achieve this behavior in EDLIF, compared to eLIF and mAdEx. One drawback of EDLIF compared to eLIF, mAdEX and Model 2, is the lack of a richer behavior repertoire at the neuron level. Despite its simplicity, EDLIF has an interesting feature that is not present in eLIF, mAdEx and Model 2. In addition to the "energetic health" parameter (K in EDLIF, $\alpha$ in eLIF and mAdEx, and $\varepsilon_p$ in Model 2), EDLIF includes a parameter controlling the sensitivity of the neuron to an energy imbalance ($\gamma$ in Eq 2). This parameter allows us to understand the impact of an energy imbalance under different neurons' sensitivity, thus it also allows us to differentiate how the same metabolic neuronal impairment (or energetic perturbation) affects two different neurons. Another distinction between EDLIF and eLIF, mAdEx, and Model 2 models, is how energy consumption is modeled. Under the eLIF, mAdEx and Model 2, a spike produces an instantaneous decrease in the available energy, thus the dynamic of spikes' energy consumption is reduced to an instant. On the other hand, under the EDLIF model, energy consumption has its own dynamic (Eqs 10 and 12). This is an important distinction because energy consumption is not an instantaneous process and, in general, the time constants associated with neuronal energy consumption are much larger than the membrane time constants, and the rate at which energy is consumed has an impact on neurons' behavior. This is particularly important when studying neuronal metabolic production impairment, as the same total energy consumption per activity could lead to different behaviors in the neuron if the rate at which that energy is consumed differs. Therefore, including energy consumption dynamics is closer to biological reality and is crucial for studying how energy dynamics affect neuronal behavior. These dynamics could also be included in the eLIF, mAdEx and Model 2 and, given their richer repertoire of behaviors, this inclusion could improve our understanding of the impact of energy dynamics on neuronal behavior.

Another limitation of our model is the inclusion of only one type of plasticity, which modifies excitatory-excitatory connections. Thus, we are neglecting other types of plasticity acting at different time scales, such as short-term plasticity or synaptic scaling. In addition, all the connections in the model, except excitatory-excitatory connections, are static. This is also an important limitation because different types of plasticity acting on other connections could have a significant impact on the network's activity and structure [51,52]. Finally, our model was not contrasted with experimental data. To the best of our knowledge, no available data simultaneously records STDP metabolism and electrophysiology. This is a relevant caveat, as the temporal dynamics of metabolism are significantly slower than those of neural activity. This kind of experimental data would be of great value in better understanding synaptic-energy dependence. To address this issue and ensure our simulations are more representative of biological conditions, we used $E_{ap} = 8.2E_{syn}$ in our simulations. This aligns our numerical experiments with the energy consumption ratio presented in [41], where the ratio between the total energy consumed by synaptic transmission and action potential generation is $C_{syn} : C_{ap} = 3$. By ensuring that our models are consistent with established energy expenditure ratios,we aim to provide more biologically relevant insights into the network dynamics.

While most neuronal and synaptic models do not account for energy dependencies, a few notable exceptions address this aspect (see [21,49,50,53–56]). Accordingly, the developed framework, analysis of dynamics, and structure in spiking neural networks under metabolic impairment, as well as the analysis of the effect of energy constraints in E-I networks, are the main contributions of this work.

Regarding future work, there are a few avenues along which to continue and extend the work presented here. Firstly, while our focus was on integrating energy-dependent long-term plasticity within excitatory-excitatory connections, other connection types could similarly undergo energy-dependent plastic alterations, warranting broader model inclusion [51]. Additionally, our emphasis on long-term synaptic modifications might benefit from an expanded scope that captures plasticity across diverse temporal scales. For instance, incorporating short-term plasticity rules, especially with insights from established models such us the formalization introduced by Tsodyks [57], could offer a richer understanding of synaptic dynamics, particularly when energy dependencies in processes like synaptic vesicle recycling are considered [9,10]. Moreover, our study's homogeneous approach, which presumed uniform neuron parameters, might be broadened to encompass the inherent variability of biological systems by investigating heterogeneous neural networks under metabolic constraints. Lastly, a more detailed exploration into the coupling phenomena among neurons, given our plasticity mechanism's reliance on spike timings, could shed light on its structural implications for networks. This would involve both analytical studies and simulations to discern the overarching effects on network architecture and function.

## Supporting information

**S1 Text**. **Supporting information for 'Unveiling the role of local metabolic constraints on the structure and activity of spiking neural networks' including stability analysis.**
(PDF)

**S2 Text**. **Supporting information for 'Unveiling the role of local metabolic constraints on the structure and activity of spiking neural networks' including average synaptic weight evolution under different parameter conditions.**
(PDF)

**S1 Table.**
(PDF)

**S2 Table.**
(PDF)

**S1 Fig.**
(TIF)

**S2 Fig.**
(TIF)

**S3 Fig.**
(TIF)

## Author contributions

**Conceptualization:** Ismael Jaras.

**Formal analysis:** Ismael Jaras, Rodrigo C. Vergara.

**Funding acquisition:** Marcos E. Orchard, Pedro E. Maldonado.

**Investigation:** Ismael Jaras, Rodrigo C. Vergara.

**Methodology:** Ismael Jaras.

**Software:** Ismael Jaras.

**Supervision:** Marcos E. Orchard, Pedro E. Maldonado, Rodrigo C. Vergara.

**Visualization:** Ismael Jaras.

**Writing – original draft:** Ismael Jaras, Rodrigo C. Vergara.

**Writing – review & editing:** Ismael Jaras, Marcos E. Orchard, Pedro E. Maldonado, Rodrigo C. Vergara.

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
