## [Decision Letter · Decision Letter 0]

11 Dec 2023

Dear Mr. Jaras,

Thank you very much for submitting your manuscript "Unveiling the role of local metabolic constraints on the structure and activity of spiking neural networks" for consideration at PLOS Computational Biology.

As with all papers reviewed by the journal, your manuscript was reviewed by members of the editorial board and by several independent reviewers. In light of the reviews (below this email), we would like to invite the resubmission of a significantly-revised version that takes into account the reviewers' comments.

All 3 reviewers agree that the research question is interesting and relevant. However, the choices and assumptions made for the neuron model, parameters, network configuration and more need to be explained better. Why was this particular setup chosen, and why is it the right one for this question? Is there a link to experimental data that justifies these choices? Secondly, the code for all the results needs to be shared, as reviewer 1 states. Thirdly, the reviewers feel that the existing literature on the topic of energy use and balanced networks needs to be integrated better in the introduction and/or discussion. Finally, the figures need some improvements (see below).

We cannot make any decision about publication until we have seen the revised manuscript and your response to the reviewers' comments. Your revised manuscript is also likely to be sent to reviewers for further evaluation.

Sincerely,

Fleur Zeldenrust

Academic Editor

PLOS Computational Biology

Marieke van Vugt

Section Editor

PLOS Computational Biology

All 3 reviewers agree that the research question is interesting and relevant. However, the choices and assumptions made for the neuron model, parameters, network configuration and more need to be explained better. Why was this particular setup chosen, and why is it the right one for this question? Is there a link to experimental data that justifies these choices? Secondly, the code for all the results needs to be shared, as reviewer 1 states. Thirdly, the reviewers feel that the existing literature on the topic of energy use and balanced networks needs to be integrated better in the introduction and/or discussion. Finally, the figures need some improvements (see below).

Reviewer's Responses to Questions

**Comments to the Authors:**

Reviewer #1: NB: For a properly formatted version of this review, see the PDF attachment.

# Review PCOMPBIOL-D-23-01765

The model and its mathematical analysis seem novel and interesting. They would indeed contribute to advancing the understanding of how metabolic constraints contribute to shaping the activity of neuronal networks.

However, a detailed discussion of the reason behind the choices made in the models, their implications in terms of biological relevance is necessary, with at the very least some justification of the relevance of the model (= neuronal dynamics + EI balanced network) to specific cases of metabolic disruption.

A justification of the values chosen for the parameters (notably \(E_{syn}\)), and their compatibility biological data on energy consumption in the brain (e.g. [Howarth2012]) must be provided.

This may bring into question the hypothesis

<math xmlns="http://www.w3.org/1998/Math/MathML"><semantics><msub><mi>E</mi><mrow><mi>a</mi><mi>p</mi></mrow></msub><msub><mover><mi>ν</mi><mo>¯</mo></mover><mi>t</mi></msub><mo>≪</mo><msub><mi>E</mi><mrow><mi>s</mi><mi>y</mi><mi>n</mi></mrow></msub><munder><mo>∑</mo><mi>k</mi></munder><msubsup><mover><mi>w</mi><mo>¯</mo></mover><mi>t</mi><mi>k</mi></msubsup><msubsup><mover><mi>ν</mi><mo>¯</mo></mover><mi>t</mi><mi>k</mi></msubsup><annotation encoding="LaTeX">E_{ap} \overline{\nu}_t \ll E_{syn} \sum_k \overline{w}_t^k \overline{\nu}^k_t </annotation></semantics></math>

A discussion about how the models fit among the landscape of other neuronal and synaptic models including energetics would also be appreciated.

In addition, the Results section needs some major changes to be more synthetic and readable. Rather than selecting seemingly arbitrary values in Figures 8 to 16, the author should provide the whole phase diagram of the network model.

The diagram should illustrate the main type of dynamics/attractors characterizing the network activity as a function of \(\gamma\), \(\eta\), and K.

This would both provide a more principled analysis of the model and enable the reader graso most of the results in a single figure.

The authors could then illustrate and discuss the different parts of the phase diagram using one figure for each type of activity (= region of phase space).

## Scientific remarks

Regarding the neuronal model : I expect the main impact of ATP deficit, associated to the Na/K-pump, to involve the resting potential (i.e. via \(g_{leak}\)), rather than \(v_r\). Though there can be changes in the reset potential during hypoxia, I am not sure: a) that this is a generic response that has been attributed directly to metabolic constraints, nor b) that it does not occur together with or later than the depolarization and with less influence on the neuronal activity (see e.g. [Müller2000] and [Chen2017]).

Of course, as the authors propose a phenomenological model, it can be fine to achieve a desired type of response to energy availability via a mechanism that differs from the biological one, but this should be explicitly mentioned and justified. This discussion was not present in the original article for the model; it would be good to have it now.

[Müller2000]: https://www.sciencedirect.com/science/article/pii/S0306452200000257?via%3Dihub#FIG1

[Chen2017]: https://www.sciencedirect.com/science/article/pii/S0014299916307890?via%3Dihub#f0010

Similarly, the choice and relevance of an all-to-all network and an EI-balanced model should be discussed. Was this done purely out of convenience in terms of mathematical analysis or in terms of the type of brain regions and metabolic issues that the authors want to model?

This is especially important as EI dynamics (notably asynchronous irregular, which is usually the one chosen for such networks) has very specific properties that have strong implications in terms of decorrelating the single cell properties and the network dynamics (see e.g. [Fardet2020]).

On the other hands, multiple types of neurodegenerative diseases are known to be associated to rhythmic activities with properties that differ drastically from the EI balanced model.

[Fardet2020]: https://journals.plos.org/ploscompbiol/article?id=10.1371/journal.pcbi.1008503

In the current state of the article, it is also unclear how important the energy in the neuronal model is (compared to the synaptic constraints) for shaping the network dynamics.

This is especially problematic since the biologically-relevant range for the parameters is not discussed and the combination of values in Figure 12 do not seem to be the most relevant for that discussion.

## Specific remarks

> Despite the significance of this relationship, specific details concerning the impact

> of metabolism on neuronal dynamics and neural network architecture remain elusive,

> creating a notable gap in the existing literature.

There are quite a few publications on this topic, particularly in the theoretical literature, that are not cited in the paper.

This sentence, and the absence of mention of these other studies, may give a distorted view of the current state of the art.

> electric activity corresponds to 92% of neuron expenses [2]

I'm not sure what part of Jolivet et al. supports that claim. In addition, "electric activity" is unclear in that context and should be replaced with something clearer.

As per Figure 2.C in [2], signaling-related energy usage is around 60% in neurons.

> Basically, realizing how crucial energy management is for shaping neural systems

That first part of the sentence should be removed.

> This study aims to advance our understanding of how local metabolic constraints

> influence the structure and activity of spiking neural networks. To achieve this, we

> introduce a novel energy-dependent plasticity rule and investigate the impact of local

> metabolic constraints through both analytical methods and numerical simulations.

This should either be moved up right after the sentences on EHP (and modified to fit within the 2nd-to-last paragraph) or removed altogether.

> Eq. 1

\(A_H\) and \(A(t)\) should be defined (homeostatic and time-dependent ATP levels).

> Figure 1

>hy aren't the firing rates in the simulations following the precise values given by the mathematical formula? Is it due to the timestep of the simulation, the number of spikes considered or some other imprecision? This should be discussed or fixed. In particular, the blue and indigo curves should, in theory, not cross in the inset.

> Eq. 5

\(\tau_{ref}\) should be defined in the text (refractory time); it might also be good to re-explicitate that the dependency in A comes from \(s_0\) (thus making it obvious that \(\tau_{ref}\) is a constant).

>Eq. 17

\(\lambda\) was already used as the firing rate in Eq. 5, please change one of the two for consistency.

> Eq. 18

\(\alpha\) was already defined as \(v_{rest} /v_{th} − 1\) in Eq. 2, please change one of the two for consistency.

> There is biological evidence suggesting the existence of excitatory-inhibitory (E-I)

> balanced networks in different parts of the brain, such as the CA3 region of the

> hippocampus, basal ganglia, and the primary visual cortex [27]

Citation [27] is a simulation study, please provide direct citations of biological papers asserting the relevance of balanced networks in each of these three regions (especially basal ganglia).

> Because both LTP and LTD depended on the activation of NMDA receptors and are absent

> in cases in which the postsynaptic neurons were GABAergic in nature [28], only

> excitatory-excitatory connections are plastic in our E-I network.

The plasticity of excitatory synapses on inhibitory postsynaptic neurons is a well-known phenomenon ([Malenka2004], [Lu2007]). It has also been demonstrated that many types of synapses can display LTP, including inhibitory ones ([Haas2006]).

[Malenka2004]: https://www.cell.com/neuron/fulltext/S0896-6273(04)00608-7

[Lu2007]: https://pubmed.ncbi.nlm.nih.gov/17804631

[Haas2006]: https://doi.org/10.1152/jn.00551.2006

> To numerically test our theoretical predictions, the network is simulated utilizing the

> Neural Simulation Tool program (NEST) [29]

Please cite the specific version of NEST that was used:

https://nest-simulator.readthedocs.io/en/stable/citing-nest.html

The Scholarpedia article is not a relevant citation for software.

> Figures 14C/15C

Do not use the average value in 15C but a 2D distribution. I would actually suggest using a distribution for Figures 8, 10, 13, and 15, both for consistency and because some other plots have values that are not completely stationary.

> In addition, following in vitro measured weights strength in neuronal cell

> assemblies [32], initial synaptic strength values for the simulated network are drawn

> from an exponential distribution

Regarding network properties ([Barral2016]) or similar experimental works in vitro or in vivo would be more relevant given the specific model studied.

[Barral2016]: https://www.nature.com/articles/nn.4415

> Eq. 40

This does not seem correct, unless I misunderstood how the matrices are constructed (in which case it should be explained in more details).

From what I understand, Eq. 39 is for a single neuron (let's call it \(i$). It could (should ?) actually be written:

<math xmlns="http://www.w3.org/1998/Math/MathML"><semantics><msubsup><mover><mi>A</mi><mo>¯</mo></mover><mi>t</mi><mi>i</mi></msubsup><mo>=</mo><msubsup><mi>A</mi><mi>H</mi><mi>i</mi></msubsup><mo>-</mo><mfrac><mn>1</mn><mi>K</mi></mfrac><mfenced close="]" open="["><mrow><msub><mi>E</mi><mrow><mi>a</mi><mi>p</mi></mrow></msub><msubsup><mover><mi>ν</mi><mo>¯</mo></mover><mi>t</mi><mi>i</mi></msubsup><mo>+</mo><msub><mi>E</mi><mrow><mi>s</mi><mi>y</mi><mi>n</mi></mrow></msub><munder><mo>∑</mo><mi>j</mi></munder><msubsup><mover><mi>w</mi><mo>¯</mo></mover><mi>t</mi><mrow><mi>i</mi><mi>j</mi></mrow></msubsup><msubsup><mover><mi>ν</mi><mo>¯</mo></mover><mi>t</mi><mi>j</mi></msubsup></mrow></mfenced><annotation encoding="LaTeX"> \overline{A}^i_t = A^i_H - \frac{1}{K} \left[ E_{ap}\overline{\nu}^i_t + E_{syn} \sum_j \overline{w}^{ij}_t \overline{\nu}^j_t \right] </annotation></semantics></math>

In that perspective, we can write the vector of activity on the set of all neurons \(i \in [1, 500]\), i.e. \(\overline{\mathbf{A}}_t = \{ A^i_t \}_i \).

The only matrix involved is \(\overline{\mathbf{w}} = \{\overline{w}^{ij}_t\}_{i, j} \) and the whole equation becomes:

<math xmlns="http://www.w3.org/1998/Math/MathML"><semantics><msub><mover><mi mathvariant="bold">A</mi><mo>¯</mo></mover><mi>t</mi></msub><mo>=</mo><msub><mi mathvariant="bold">A</mi><mi>H</mi></msub><mo>-</mo><mfrac><mn>1</mn><mi>K</mi></mfrac><mfenced close="]" open="["><mrow><msub><mi>E</mi><mrow><mi>a</mi><mi>p</mi></mrow></msub><msub><mover><mi mathvariant="bold-italic">ν</mi><mo>¯</mo></mover><mi>t</mi></msub><mo>+</mo><msub><mi>E</mi><mrow><mi>s</mi><mi>y</mi><mi>n</mi></mrow></msub><msub><mover><mi mathvariant="bold">w</mi><mo>¯</mo></mover><mi>t</mi></msub><msub><mover><mi mathvariant="bold-italic">ν</mi><mo>¯</mo></mover><mi>t</mi></msub></mrow></mfenced><annotation encoding="LaTeX"> \overline{\mathbf{A}}_t = \mathbf{A}_H - \frac{1}{K} \left[ E_{ap}\overline{\boldsymbol{\nu}}_t + E_{syn} \overline{\mathbf{w}}_t \overline{\boldsymbol{\nu}}_t \right] </annotation></semantics></math>

I.e.

<math xmlns="http://www.w3.org/1998/Math/MathML"><semantics><msub><mover><mi mathvariant="bold">A</mi><mo>¯</mo></mover><mi>t</mi></msub><mo>=</mo><msub><mi mathvariant="bold">A</mi><mi>H</mi></msub><mo>-</mo><mfrac><mn>1</mn><mi>K</mi></mfrac><mfenced close="]" open="["><mrow><msub><mi>E</mi><mrow><mi>a</mi><mi>p</mi></mrow></msub><mi mathvariant="normal"></mi><mo>+</mo><msub><mi>E</mi><mrow><mi>s</mi><mi>y</mi><mi>n</mi></mrow></msub><msub><mover><mi mathvariant="bold">w</mi><mo>¯</mo></mover><mi>t</mi></msub></mrow></mfenced><msub><mover><mi mathvariant="bold-italic">ν</mi><mo>¯</mo></mover><mi>t</mi></msub><annotation encoding="LaTeX"> \overline{\mathbf{A}}_t = \mathbf{A}_H - \frac{1}{K} \left[ E_{ap} \mathbb{I} + E_{syn} \overline{\mathbf{w}}_t \right] \overline{\boldsymbol{\nu}}_t </annotation></semantics></math>

Or, if one wants to transpose everything,

<math xmlns="http://www.w3.org/1998/Math/MathML"><semantics><msubsup><mover><mi mathvariant="bold">A</mi><mo>¯</mo></mover><mi>t</mi><mo>⊤</mo></msubsup><mo>=</mo><msubsup><mi mathvariant="bold">A</mi><mi>H</mi><mo>⊤</mo></msubsup><mo>-</mo><mfrac><mn>1</mn><mi>K</mi></mfrac><msubsup><mover><mi mathvariant="bold-italic">ν</mi><mo>¯</mo></mover><mi>t</mi><mo>⊤</mo></msubsup><mfenced close="]" open="["><mrow><msub><mi>E</mi><mrow><mi>a</mi><mi>p</mi></mrow></msub><mi mathvariant="normal"></mi><mo>+</mo><msub><mi>E</mi><mrow><mi>s</mi><mi>y</mi><mi>n</mi></mrow></msub><msubsup><mover><mi mathvariant="bold">w</mi><mo>¯</mo></mover><mi>t</mi><mo>⊤</mo></msubsup></mrow></mfenced><annotation encoding="LaTeX"> \overline{\mathbf{A}}_t^\top = \mathbf{A}_H^\top - \frac{1}{K} \overline{\boldsymbol{\nu}}_t^\top \left[ E_{ap} \mathbb{I} + E_{syn} \overline{\mathbf{w}}_t^\top \right] </annotation></semantics></math>

> it is possible to neglect the neuron’s own action potentials energy consumption

This is indeed the case with the values of \(E_{ap}\) and \(E_{syn}\). However, it seems to directly contradict the results from [2] and ([Howarth2012]) where \(E_{ap} / E_{syn}\) is 20%/60%.

As the values of \(E_{ap}\) and \(E_{syn}\) are currently not justified, it seems absolutely necessary to explain why these specific values where chosen.

[Howarth2012]: http://journals.sagepub.com/doi/10.1038/jcbfm.2012.35

> Exploring synaptic energy imbalances sensitivity

These paragraphs are really hard to read, even knowing what is going on. An additional schematics of the transition, rather than lengthy paragraphs, would probably make this much more understandable.

> Interestingly, when the synapses are highly sensitive to energy

> imbalance (Fig. 10E and 10F), it is not possible for all the neurons

> to achieve the metabolic fixed point, and some neurons achieve

> the silent fixed point previously described.

A more precise discussion of the mathematical transition for silenced neurons would be appreciated.

> Fig 9C (and more generally)

Mean and standard deviation are not relevant for a bimodal distribution, please plot each monomodal subpopulation separately.

## Technical remarks

The provided code contains only the NESTML model

([https://github.com/Wiss/nestml](https://github.com/Wiss/nestml)).

It would be preferable to provide all the relevant code necessary to replicate the result in the study. Furthermore, rather than include the new model within a fork of the nestml repository (which requires to download 80MB of data to access 2 files of 3kB each), it would be better to include them in a separate repositories.

I suggest uploading all the relevant code in the following form:

```

Jaras_et_al_MetabolicConstraintsModel/

├── README.md # instructions to install the models and run the scripts

├── Models/

│ ├── ed_stdp_synapse.nestml

│ └── third_factor_stdp_synapse.nestml

└── Simulations/

├── fig1.py

├── ...

└── fig16.py

```

This is just an example structure. Some figures may be grouped within a single file and some separate simulation files may be added to separate simulations and analysis whenever relevant.

## Typographic remarks

Adapt the size of figures or the figures' fontsize such that the text in the figures appears similar to the main text.

There are issues with quotes. If LaTeX was used, make sure to use ``this'' and not "that" for quotes.

> where synaptic potentiation and depression balance each other in average.

*on* average

Reviewer #2: In this paper the authors sought to explore the effects of modifying a spike-timing-dependent plasticity rule to account for energy constraints on a neuron. They analyzed in particular a spiking neural network’s ability to attain a fixed mean firing rate as a function of various changes to the metabolic parametrization. They further considered pathological effects on a fixed firing rate due to metabolic limitations. I agree with the authors that energetics are not (to my knowledge) considered in most neural network studies, and in that regard this work could fill a gap in the literature. However, a number of concerns and limitations in the present work dampen my enthusiasm.

1. The authors claim to explore E/I balanced networks yet neglect an entire field’s worth of work on the subject (see, e.g. van Vreeswijk, Sompolinsky, Renart, Doiron) demonstrating a rich variety of dynamics which could have been explored. Nor do the authors defend this choice through an appeal to other work which discusses different notions of balance (e.g. Ahmadian & Miller 2021). Figures 3 does not add much to the story outside of illustrating parameters. Moreover, these weight parameters which are critical in appreciating balance and the dynamical regime of the network are nowhere provided (and not included in Table 1). It would be important to include the chosen w_ij values, and the range of w_ee, or max(w_ee) (this I attempted to infer from later plots to be O(1)). Additionally Figure 4 is at best not helpful and at worst misleading because the network is stated to be all-to-all connected and no concept of distance-dependent connectivity or weight modulation is mentioned, so the spatial distribution of the neurons on some grid is irrelevant. Further, balanced regimes can lead to chaotic dynamics. While the firing rate plots demonstrate this is not the case here, all of the math related to a fixed point solution depends on assumptions about the dynamical regime of the network which is not stated.

2. I am confused about the author’s approach in comparing theory and simulations. Much biological richness was omitted in choosing homogeneous network parameters. More complete analysis could perhaps have been achieved with a smaller, simplified architecture (e.g. two E neurons). Further, the authors comment on numerous occasions that certain approximations hold in a large-N limit (e.g. pg 16 last paragraph). In this vein, no attempt to match theory and simulation outside of qualitative discussion was attempted for e.g. figures 10, 13, 15 which seem to permit such a representation. Further, the figures in which some theoretical-type estimate was included didn’t have any mention of why the sims would (and often do) disagree.

3. In the Discussion, I think the suggestion of therapeutic relevance is a bit far-fetched. I’m also not convinced that the suppression of a subset of the neuronal activity through what I understand to be disconnection (or extremely low E-E connections) is consistent with the idea of sparsely-firing neurons in cortex, as suggested in the discussion. More would need to be said/done to further this case.

4. The network effects seem rather insensitive to choice of \gamma (Figs 1 and 9). Also, no discussion on the relevant order of certain parameters was attempted. This could help mathematically, but in addition the authors claimed to capture existing results from data yet this does not seem to be seriously considered without such a discussion.

5. The only really interesting dynamics appear only in the (apparently) pathological case of K very small (Fig. 14). What happens in the case of heterogeneous networks? Currently the only source of heterogeneity is in the input.

6. K is overloaded as a parameter in the text (eqs 7 and 17).

Additional points:

1. Eq 1, I(t) is not defined. Should be stated.

2. Eq 2, A_H is not clearly defined until later, and should be discussed at least within the same section.

3. Between eq’s 10 and 11; 12 and 13 it is unclear why a Heaviside is introduced when the integral over epsilon is already bounded to 1, and then an exponential function is later used for epsilon. It seems to add a number of equations and text which seem to be unnecessary.

4. Following Eq 20, it was stated that Eq 20 was left as-is for future general readers while a specific case is examined here which dramatically simplifies the \Delta w equation. Considering a general form is already provided in eq 17, it would be nicer to simply write the specific case examined here in place of the current Eq 20.

5. Should eq 27 not be a strict equality in the linear case? What is approximated?

6. The extensive discussion of trivial solutions (e.g. pg 16 of the manuscript) detracts more than it adds from the exposition.

7. Eq 4 V_th should be lower-case

8. Figure 11 does not illustrate the two subpopulations, simply shows a separation in the ATP dynamics. Would be better to show the actual network responses as well.

9. The notation for total presynaptic input is awkward and should be reconsidered (i.e. superscript ex \to ex, k is very cumbersome)

Reviewer #3: This study presents an energy-dependent STDP plasticity rule combined with a previously established energy-dependent LIF model to investigate the effects of metabolic conditions on firing in a spiking neural network model. The strength of the study is its focus on the understudied topic of energetics in neural network models as well as its employment of both analytical and numerical methods. Their simulations predict network activity affected by metabolic imbalances and disruptions.

This is a potentially highly interesting energy-dependent model of synaptic plasticity. However, the authors do not show how close it is to real neurons and synapses in terms of the time scales of ATP and synaptic changes. The authors have not tuned the model to experimental data. They only claim that the model replicates the "suppression of long-term potentiation when the available energy level on the postsynaptic neuron drops". It would be important to show that the simulated weight changes, FR changes and energy expenditure/supply changes (ideally under different metabolic conditions) correspond to some measured weight and FR (and ideally also ATP) changes. Models published in Plos CB are usually closely linked to experimental data. If the authors cannot demonstrate a strong link to data, then the manucript - although topical and exciting - should be submitted to a more theoretical (i.e. less data-driven) computational journal.

In sumary, the main issue is the lack of links to data to tune or justify the models.

Here are more specific points:

* The authors should clarify which experimental data justify that the "energy-dependent STDP rule uses postsynaptic available energy as a neuromodulator, resulting in the suppression of LTP and enhancement of LTD at low postsynaptic energy levels (Fig. 5)

* Why is everything exponentially dependent on ATP level? What is a reasonable level of 'sensitivity'? Looking at, for example, Fig 10, everything seems to be in a very small range of A_t.

* Some missing context (that makes the problem of links to data worse) and missing citations of previous literature. For example: Smith et al (Neuro Behav Rev, 2011) on human learning with metabolic differences, Padamsey et al (Neuron, 2022) on changing neuronal activity and behaviour under metabolic restrictions, and Pache & van Rossum (Curr Opp Neurobiol, 2023) on learning rules that reduce energetic demands (also the paper from Li & van Rossum (Elife 2020) is missing). For more general context on the trade-offs between energy and neuronal/synaptic function in the context of optimality see e.g. Harris et al. (Curr Biol. 2015), Harris et al. (PLoS CB 2019), Mahajan and Nadkarni (eNeuro 2020), Jedlicka et al. (Open Biol 2022).Importantly, the authors should definitely mention papers specifically focused on synaptic plasticity and energetics from Jan Karbowski (J Neurophysiol 2019, J Comput Neurosci 2021) and also a previously published energy-related neuronal firing model from Fardet & Levina (Plos CB 2020).

* Additive learning rules. There is some evidence that the skewed distributions of synaptic weights produced by multiplicative learning might be more energetically efficient (ie Goetz at al (PNAS, 2021), see also Roessler et al (Open Biol, 2023), Eggl et al (Comm Biol 2023). This might strongly impact the results given the interplay between energy consumption and learning. This is not meant as a major point but rather as a caveat that could be mentioned in the Discussion.

* Figure 1 shows that reduced energetic availability increases the firing rate due to the inability of the cells to properly reset. In later results sections (ie page 20), it is assumed that reduced energetic availability reduces the firing rate (and not just through reduced LTP). How are these two phenomena to be reconciled?

* Much of the results are interpreted under the idea of 'balanced' ATP production and consumption, but the consequences of excess ATP accumulation are not really explored. It seems that it might lead to over-potentiation and instability, but it would be good to see this.

* I don't understand what Fig 4 shows. Are synaptic connectivities or strengths distant dependent? This does not seem to be the case in model.

* Short-term presynaptic plasticity was not explored although the presynaptic energy/calcium/synaptic vesicle pool would affect the firing rates on the time scale that has been simulated in the paper (see e.g. Mahajan & Nadkarni, eNeuro 2020)

* "experimentally, it is difficult to precisely measure ATP concentrations in single neurons or populations. To overcome this difficulty and to define a more general framework, here we will use percentage units to quantify available energy" That is fine but how was the value for the time constant tau_A determined? "K = 1/τA is the rate at which ATP can be produced in the neuron. Again, how is the model related to the data?

* The authors write: "Energy Homeostasis Principle (EHP) was introduced in [10, 11]. The rationale behind this principle posits that neurons, in meeting their metabolic needs, inadvertently address behavioral challenges as an epiphenomenon." Can the authors explain this sentence? Considering" behavioral challenges" as an epiphenomenon does not seem plausible. This should be make clearer or changed.

Minor:

* Reference 23 contains first, not last names.

**Have the authors made all data and (if applicable) computational code underlying the findings in their manuscript fully available?**

Reviewer #1: **No: **The provided code contains only the NESTML model

([https://github.com/Wiss/nestml](https://github.com/Wiss/nestml)).

It would be preferable to provide all the relevant code necessary to replicate the result in the study. Furthermore, rather than include the new model within a fork of the nestml repository (which requires to download 80MB of data to access 2 files of 3kB each), it would be better to include them in a separate repositories.

I suggest uploading all the relevant code in the following form:

```

Jaras_et_al_MetabolicConstraintsModel/

├── README.md # instructions to install the models and run the scripts

├── Models/

│ ├── ed_stdp_synapse.nestml

│ └── third_factor_stdp_synapse.nestml

└── Simulations/

├── fig1.py

├── ...

└── fig16.py

```

This is just an example structure. Some figures may be grouped within a single file and some separate simulation files may be added to separate simulations and analysis whenever relevant.

Reviewer #2: Yes

Reviewer #3: Yes

PLOS authors have the option to publish the peer review history of their article (what does this mean?). If published, this will include your full peer review and any attached files.

Reviewer #1: **Yes: **Tanguy Fardet

Reviewer #2: No

Reviewer #3: No
---

## [Decision Letter · Decision Letter 1]

10 Sep 2024

Dear Mr. Jaras,

Thank you very much for submitting your manuscript "Unveiling the role of local metabolic constraints on the structure and activity of spiking neural networks" for consideration at PLOS Computational Biology.

As with all papers reviewed by the journal, your manuscript was reviewed by members of the editorial board and by several independent reviewers. In light of the reviews (below this email), we would like to invite the resubmission of a significantly-revised version that takes into account the reviewers' comments.

Although all three reviewers feel that the manuscript has improved considerably, reviewers 1 and 2 still feel that there is still quite some improvement possible. Some claims and assumptions need to be addressed more thoroughly, and some figures need some editing.

We cannot make any decision about publication until we have seen the revised manuscript and your response to the reviewers' comments. Your revised manuscript is also likely to be sent to reviewers for further evaluation.

Sincerely,

Fleur Zeldenrust

Academic Editor

PLOS Computational Biology

Marieke van Vugt

Section Editor

PLOS Computational Biology

Although all three reviewers feel that the manuscript has improved considerably, reviewers 1 and 2 still feel that there is still quite some improvement possible. Some claims and assumptions need to be addressed more thoroughly, and some figures need some editing.

Reviewer's Responses to Questions

**Comments to the Authors:**

Reviewer #1: NB: For a properly formatted version of this review, see the PDF attachment.

All line numbers refer to the new manuscript, not the diff file.

# Review PCOMPBIOL-D-23-01765 (R2)

This revised version of the manuscript is a significant improvement over the previous version.

Though I maintain that a proper analysis of the phase space and fixed points stability would have further improved the article, I think that the manuscript can be published close to its current state.

However, before the manuscript can be published, a few key issues, detailed below, must still be addressed.

## Comments on the authors' replies

### Comment 10

> Given that the reviewer has not given any concrete reason to perform such change, and that we disagree on the suggestion, we have not made the requested change. The aim of the studies is usually left by the end of the introduction, as we did it. We also think that the objective is properly described. Therefore we were not able to understand the rationale of the suggested change.

The aim usually comes before the summary of the content; here it seems to be split into two by the paragraph starting by "Through our mathematical analysis...", which feels strange to me. But this is indeed a matter of personal preference.

### Distorted view of the field

> The discussion has been updated including references for the eLIF and mAdEx models, as well as an explanation of our decision for using the EDLIF model.

I appreciate the addition; however, my remark about the manuscript giving a distorted view of the field still stands.

In particular, there is another phenomenomenological model from [Chhabria2016] that is quite similar (modeling ATP concentration) and has been used to model the influence of energy on network dynamics, so a more general comparison of the results to these models would also be relevant.

Finally, as the authors also cite conductance-based models from Le Masson and Jolivet, I am unsure why other works from e.g. Knowlton or Ullah, Wei and Schiff are not included, especially in the discussion about neurodegenerative diseases.

[Chhabria2016]: https://www.frontiersin.org/articles/10.3389/fneur.2016.00024/full

### Neuronal cultures

> a metabolic view of neural network activity is easier to assess in vitro using multielectrode arrays [where] it is reasonable to suppose that neurons present the possibility of a saturated network with an all-to-all connectivity.

Even for 500 neurons, **cultured networks do not display all-to-all connectivity**.

The typical connectivity is quite sparse, with around 100 connected neighbours per neurons [Soriano2008], [Barral2016].

The cited paper [39] refers to functional connectivity, not physical connectivity.

As I mentioned before, it is perfectly understandable for the authors to use an all-to-all connectivity to simplify the link between the mathematical analysis and the simulations, but they should not imply that this type of connectivity corresponds to standard *in vitro* structures.

[Soriano2008]: https://www.ncbi.nlm.nih.gov/pmc/articles/PMC2544527/

[Barral2016]: https://www.nature.com/articles/nn.4415

## Scientific remarks

### Neglecting action potential energy consumption and energy relationship (Comment 22 and L 663)

The authors seem to have missed or forgotten one of the points of my comment 22 in the previous round, which was about the derivation of equations 41 and 42 by neglecting the action potential energy consumption.

I maintain that, given that the premise seems incorrect, these equations do not appear relevant and that the following statement, on line 663, is probably incorrect:

> both simulations satisfy the energy relationship defined by Eq. 42.

This equation relies on the hypothesis that $E_{ap}\overline{\nu} \ll E_{syn} \sum \overline{w} \overline{\nu}$.

The experimental reports we have for both values give $A_{syn} \approx 3 A_{ap}$, which the authors have acknowledged in response to my initial comment.

In addition, the values given in figure 7 c and d are :

- $A_{ap} = E_{ap}\overline{\nu} \sim 4.5 \times 100 = 450$

- $A_{syn} = E_{syn} \sum \overline{w} \overline{\nu} \sim 0.5 \times 1700 = 850$

giving a ratio lower than 2, which does not satisfy the hypothesis leading to Eq. 42.

I don't think this is a major issue for the paper as a whole because this does not seem to be used anywhere (as far as I can tell), but if it is indeed incorrect, it should be removed.

### Oscillations (L 640)

The theoretical prediction for increased oscillations (point 3) feels rather hand-wavy though there are indeed reason to expect increased oscillations. The explanations given from line 715 are much better in that regard and should be properly summarized in point 3 to replace the current phrasing. Ideally, the stability of the fixed point should have been analyzed to show if lower values of $K$ lead to a phase transition from a stable to an unstable (saddle?) fixed point.

Figure 9 even suggest that the attractor dynamics may involve spiral fixed-points or more complex structures.

### Comparison of $\eta$ vs $\gamma$ (L 798)

> In particular, the synaptic sensitivity ($\eta$) has a greater impact than the neuronal sensitivity ($\gamma$) on the network dynamics and fixed point.

I do not think that the current results are sufficient to back that claim.

In particular, though both variables appear in similar exponential terms, the range explored was not the same for both ($\eta$ was set up to 100 while $\gamma$ was only set up to 50). At the very least, the authors need to explore the same range for both variables before making that claim.

Though we know that changes in $\gamma$ do not change the position of the fixed point, they may change its stability.

To prove the initial claim, the authors would need to demonstrate that changes in $\eta$ lead to several phase transitions or quantitative changes which cannot be achieved by changing $\gamma$, or that these changes would appear for unrealistic values of $\gamma$. As the analysis of phase-space and the fixed-point stability has not been performed, this claim cannot be made at the moment.

## General remarks

Both K and $\tau_A = 1/K$ are used, which seems to provide unnecessary confusion. Unless there is a specific reason to keep both, please choose either one or the other throughout the text.

The use of the capital letter $A$ to designate both energy (ATP) levels and energy consumption ($A_{ap}$, $A_{syn}$) is probably not the best choice. I would suggest using $A$ only for the ATP level and replacing $A_{ap/syn}$ by another letter (e.g. $c$) and using $C(t) = \int c(t')dt'$ in equation 16.

It would be useful to adapt figures 6c, 7c, and 8c to test their overlap with the theoretical nullclines from figure 4.

This would enable us to see how well the theoretical predictions fit the simulation results.

## Specific remarks

L 166 : there is an ambiguity for the overline notation that usually means "average" (which is also used later multiple times with that meaning, e.g. for $\overline{\nu}$), I would suggest to use another notation, e.g. $\tilde{w}$.

L 177 and L 187 : replace "For finding" by "To find".

Eq 14 : replace "spike s" by "spike f" to be consistent with eq. 16?

Eq 16 :

- the terms are the integrals of the action potential and the synaptic consumption rates, not the rates themselves, so $A_{ap}$ and $A_{syn}$ should not be used together with the braces; use something like "action potential contribution" and "synaptic contribution" instead, or define $C_{ap}$ and $C_{syn}$ and use these.

- remove the Heaviside function from the action potential contribution, as it should have been integrated away.

L 218 :

> This same mechanism has been linked to plasticity [29], where they were able to show that energetic stress can suppress LTP, while rescuing it by removing the energetic stress.

The end of the sentence is not clear to me, from article [29] I believe the sentence should read "while treatments targeting energetic pathways can rescue it by removing the energetic stress."

L 270 : the last value should be $n_I$ and not $n_{in}$.

**Results :**

* Default values for parameters $\gamma$, $\eta$, $K$ should be provided in a 2nd table in the supplementary material so we know which values are used when the other parameters are changed.

At the moment, they are mostly specified in the main text or the figure caption, but some values are missing for specific simulations (notably K).

* L 517 :

- $n_{ex}$ should be $n_E$

- why is $\nu_{max}$ given as 0.125 and not 125, since the unit is Hz?

- $A_c$ is an energy consumption rate so its unit should be homogeneous to $\%/s$

* Figures 6, 7, 8, please add a label stating that the colorbar give the average ATP level in the steady-state (or explain what it is if it is something else). I would also suggest to not fill the symbols in the legend for panels c. For figure 8, please rescale the colorbar to make the differences visible (if it is indeed the average value, I imagine it should not go below 96).

**Discussion :**

L 855 :

> neurodegeneration has been treated by increasing mitochondrial energy production or affecting activity through neurotransmission (e.g., pregabalin to inhibit or levodopa to increase activity)

the use of levodopa has more complex results than simply to "increase activity": in the basal ganglia, dopamine either excites or inhibits inhibitory neurons that will themselves inhibit the activity of other neurons.

L 860 :

> general terms,as currently done

missing whitespace

L 862 :

> Neurons are highly oxidative and selectively low-tolerant to oxidative stress [44].

I don't think this sentence is correct in English and would suggest rephrasing to something like "and some neuronal populations can be vulnerable to oxidative stress".

L 863 :

> They maintain a constant expenditure regardless of activity level [45].

I don't think this corresponds to the current consensus on neuronal metabolims, see e.g. [Watts2018].

Furthermore, [45] states that the NKP may drive mitochondrial activity, not that the energy expenditure is constant.

[Watts2018]: https://www.frontiersin.org/journals/molecular-neuroscience/articles/10.3389/fnmol.2018.00216/full

L 923 :

> the ratio between total energy consumed by action potential and synapses is 60 : 20

This sentence is misleading as it seems like 60 corresponds to the constribution of the action potential. Please rephrase it into something like "the ratio between the total energy consumed by synaptic transmission and the energy consumed by action potential generation is $C_{syn} / C_{ap} = 3$."

L 926 :

> In general, previous neuronal and synaptic models do not account for energy dependencies (some exception can be found in [21, 49, 53]).

while it is true that most publications in computational neuroscience do not consider energetic constraints, the part in parenthesis again provides a distorted image of the field.

There are phenomenological models on the subject dating back to 2016 and, if one also considers conductance-based models (which the authors apparently do given that they cite [21] and [53]), there are many more papers specifically dedicated to the subject, notably from the authors mentioned in my first comment, some of them discussing a phase-space analysis of their models [Knowlton2018], [Yao2018].

[Knowlton2018]: http://www.physiology.org/doi/10.1152/jn.00351.2017

[Yao2018]: http://aip.scitation.org/doi/10.1063/1.5018707

**Code**

this is unrelated to the current paper, but, since I was now able to test the code, I noticed that it uses weighted clustering coefficients. I suggest having a look at these articles if there are plans to use weighted clustering in future work, to make sure that the definition chosen is indeed appropriate given the scientific question:

- [10.1103/PhysRevResearch.3.043124](https://journals.aps.org/prresearch/abstract/10.1103/PhysRevResearch.3.043124)

- [https://hal.science/hal-04427288](https://hal.science/hal-04427288)

Reviewer #2: Review:

Upon review of the updated article, I still have many concerns.

Response to Comment 30.

I’m afraid this is not accurate. See, for example, Akil et al. 2021 PloS Comp Bio, in which they study STDP in balanced spiking networks. “E/I balanced networks” has a particular connotation in terms of the parametric regime regarding synaptic weights, which may be assumed fixed, or plastic. Since this is not what this paper is addressing (clearly, from Fig. 5, in which the ‘non-metabolic’ situation explodes except for refractory effects), lines 263/264 (and other instances) should be modified to omit the term “balanced”.

A more general related concern is what the role of inhibition in this model is, at all. From Fig 5, the net inhibitory current is orders of magnitude below that of excitation, suggesting that inhibition is really having little to no effect on the excitatory dynamics. This then begs the question, as to whether the ‘metabolic’ effects are justifiably the main actors, and what, aside from the silencing of some neurons in Fig 6, is the role of inhibition here? I believe that in reality, they are not doing much, and some support for this is the explanatory figure attached following Comment 31 in the response, where an excitatory-only network was simulated. In general, how robust is this, if inhibition is removed? It may be justified to include a simulated excitatory-only network in the text for comparison.

Response to Comment 31.

I apologize that my point was not well made, given the author’s response. To clarify, I was suggesting that a simplified model could do just as good as a homogeneous one in some cases, with the benefit of explaining some of the more complex oscillatory behavior or the like. Just a thought.

Importantly, a significant omission in the main text are any plots showing the evolution of the E-E weights, which were shown in the rebuttal in an explanatory fashion. I think weight evolutions plots should be included in the main text, and amended to better illustrate the claims in the main text. This is especially true considering that the weights appear in only one case (\eta = 0) to have actually stabilized.

Response to Comment 36.

The Heaviside function is still present from Eq 10 on, even though the integral of \epsilon from 0 to \infty takes care of this. Alternatively, if the constraint on the epsilons is relaxed to -infty to infty then the Heaviside is fine.

Further concerns:

1. It would be nice to have some discussion of the shape of the rate to A curve (eq. 5), which shows that more ATP leads to reduced firing (e.g. from Jaras et al 2021). Also, some discussion of the locality of this solution, since in the A \to 0 limit there would still be a positive firing rate. Further, recent work from Chintaluri & Vogels (2023) suggests that increased ATP should lead to spike discharge, which would imply the opposite trend of this curve.

2. Given your definitions for w and v, eq. 24 is not correct, since the dimensions are mismatched. Should be either v*w or w should be defined as [w_ee, w_ei; w_ie, w_ii].

3. Lines 323/324 (after eq 27). You state that one needs constant weights for constant firing rates. This isn’t correct - you’ve concluded you need constant weights in expectation.

4. Line 483 and on… could be illustrated in Figure 4 better to help the reader. Is Fig. 4 the actual solution to an instantiation of the equations or just an illustration? Line 530 and the caption suggest it’s an illustration. If it’s not, the parameters which were used to compute it should be provided.

5. A key point the authors emphasize is the silent neural population that emerges. However, no robustness arguments are explored here. How robust is this to changes in the initial conditions? Mean of the exponential distribution? Or, different distributions altogether e.g. gaussians with different means and variances? In a related vein:

-Line 553 and Fig 6. These oscillations appear to sample the firing rate floor. Are more and more neurons falling silent each time until enough are dead that the remainder stabilize at some equilibrium? Line 574 is also incomplete. Why is their energy consumption not zero? Because of incoming activity? Please clarify.

-Line 581. “Silent neurons emerge as a consequence of energy constraints”. Is this true in general? How sensitive is this to the initial weight distribution? If the initial E-E weights are weaker (or, the I to E weights stronger) is this still the case?

6. Fig 6. It appears that the blue (\eta = 30) population is also separating to some extent, especially in the E-E weights. If the simulation was run longer would they continue to diverge?

7. Figure 7d and similar (e.g. Fig 8d): plot the theoretical constraint line(s) from eq. 42 to demonstrate the accuracy of the result. Otherwise, the statement in line 613 is not so clear, just that they covary.

8. Figs 7/8. It is virtually impossible to see the color of the points on the current plots. If this is important to show together with the net currents, please find another way to plot the effect. And, please label the color bar on the plot.

Additional points:

1. A_B is used for both basal consumption and basal production. Unless these are always the same (in which case it should be justified) these should be denoted differently.

2. Line 373 the upper bound should be exp(-\eta) not 1.

3. Line 519. But what, then, is a reasonable E_{syn}?

4. Line 670. I don’t find this very clear to “contrast”; neither scrolling back and forth, nor the clear differences in the axis scales. If this is important, please find a clearer way to show it with a plot.

5. Line 383 this ‘potentiation-depression loop’ point is not well-illustrated in Fig 2, as suggested by the parenthetical. The figure could be updated to make this clearer.

6. I find the second paragraph of the introduction (lines 12 - 25) does not add anything of interest. I don’t think anyone believes that energy management is not important for normal brain function. Neuronal activity drives behavior, and requires energy to do so. So fMRI is correlated with energetics by definition.

7. Section beginning on Line 497. Fig 5 panels b,c are not discussed. Given that they show up later, they should be introduced better. Please label the color bar. How sensitive are these results to the choice of time averaging at the end of the simulation?

8. Line 819. I don’t think this is a compelling argument for why certain neurons fire so sparsely (or, here, not at all). Indeed, in the brain there is also ample inhibitory plasticity which could jointly adjust various set points. Further, despite it being a common assumption in models, a sustained 100Hz firing rate fixed point is not the goal of neural circuits. It also wouldn’t explain why those neurons exist in the first place, and from a neural coding perspective leaves a lot to explain.

Reviewer #3: The authors have generally responded well to my comments and I believe that the paper would be suitable for publication if the other reviewers believe that their comments have been addressed. The new manuscript makes the assumptions behind the model much clearer in the cases where data is limited or unavailable.

**Have the authors made all data and (if applicable) computational code underlying the findings in their manuscript fully available?**

Reviewer #1: Yes

Reviewer #2: Yes

Reviewer #3: Yes

PLOS authors have the option to publish the peer review history of their article (what does this mean?). If published, this will include your full peer review and any attached files.

Reviewer #1: **Yes: **Tanguy Fardet

Reviewer #2: No

Reviewer #3: No
---

## [Decision Letter · Decision Letter 2]

3 Feb 2025

PCOMPBIOL-D-23-01765R2

Unveiling the role of local metabolic constraints on the structure and activity of spiking neural networks

PLOS Computational Biology

Dear Dr. Jaras,

Thank you for submitting your manuscript to PLOS Computational Biology. After careful consideration, we feel that it has merit but does not fully meet PLOS Computational Biology's publication criteria as it currently stands. Therefore, we invite you to submit a revised version of the manuscript that addresses the points raised during the review process.

Please submit your revised manuscript within 30 days Apr 05 2025 11:59PM. If you will need more time than this to complete your revisions, please reply to this message or contact the journal office at ploscompbiol@plos.org. Please include the following items when submitting your revised manuscript:

We look forward to receiving your revised manuscript.

Kind regards,

Fleur Zeldenrust

Academic Editor

PLOS Computational Biology

Marieke van Vugt

Section Editor

PLOS Computational Biology

**Additional Editor Comments :**

All reviewers agree that the paper has improved significantly. Reviewer 2 mentions a few minor issues, and reviewer 1 mentions a few typographic issues, that in my opinion could be easily solved.

**Journal Requirements:**

1) Please include the affiliation of Ismael Jaras in the online submission form and ensure that it matches the affiliation listed on the manuscript title page.

**Reviewers' comments:**

Reviewer's Responses to Questions

**Comments to the Authors:**

**Please note that one of the reviews is uploaded as an attachment.**

Reviewer #1: NB: For a properly formatted version of this review, see the PDF attachment.

# Review PCOMPBIOL-D-23-01765_R2

This new revision further improved the quality of the article and answered my previous reservations.

I think this work will prove to be a valuable contribution to the field of neuronal metabolism.

Apart from minor typographic issues (that can be addressed without the need for another review), I consider that the article can be published as is.

## Minor remark

> To our knowledge, they maintain a constant availability regardless ofactivity level [45] when using physiological-like stimulation.

"physiological-like" was not clear to me before I read the reply to my comments.

Maybe something like "when applying stimulations that do not exceed physiological levels" would be clearer?

## Typographic remarks

The firing rate vector $\boldsymbol{\overline{\nu}}_t$ (Eq. 24 to 27 and in Eq. 38) should be made **bold** using ``\boldsymbol`` to show that it is a vector, according to the authors' typographic choice.

Lines 339-340 should read: "the network architecture is define**d**"

Line 858, it should be "levodopa" and not "levopoda".

Reviewer #2: Response to Comment 25.

I’m not convinced. Fig 5 shows that in the ‘base’ case, inhibitory inputs are two orders of magnitude below excitation, but the responses saturate due to refractory period alone, and the energetics stabilize. Hence it seems that the energy constraints are doing the work of stabilizing the system. In principle there is nothing “special” about the inhibitory term in eq. 41; so if one were to split a purely excitatory population in two with one subset having plastic and the other non-plastic weights, you could play the same game as in eq’s 40 & 41 and separate the non-plastic and plastic subsets. Further, I understand the thought experiment in the rebuttal, but it seems deeply contrived. For example, if one considers assemblies of neurons in cortex, E and I neurons may strongly co-cluster. By the same logic, assuming strong feedforward drive to both E and I, I neurons would cause the strong assembly connections to weaken. If one believes in, e.g. assemblies as storing memories, this would seem to be a problem.

Response to Comment 26.

Given that this is a paper about the role of plasticity, this doesn’t seem to be a good rebuttal. I think it would be of help to the reader to see the weights evolving in time, akin to the firing rates. Reviewer 1 made a related request in their Comment 8 regarding plotting theoretical predictions on figures 6c/7c/8c. This hearkens back to my Comment 31 regarding Fig 4 as a schematic.. should Fig 4 not be computable for stated parameters values? Then figs 6c/7c/8c should be able to have simulations and theory agree.

Response to Comment 27.

re: Heaviside. Sorry, my confusion was unrelated; I had made a mistake. But I agree, it’s ok.

Response to Comment 28.

re: solution locality (i.e. A \to 0 limit). I disagree; and I think a brief statement of this would make sense e.g. around Line 515 where you mention this limit, to insure the reader knows that you’re excluding it.

Response to Comment 37.

Yes, sorry, another mistake on my end; it’s fine as-is.

Reviewer #3: The paper can be accepted if the two other reviewers agree.

**Have the authors made all data and (if applicable) computational code underlying the findings in their manuscript fully available?**

Reviewer #1: Yes

Reviewer #2: Yes

Reviewer #3: Yes

PLOS authors have the option to publish the peer review history of their article (what does this mean?). If published, this will include your full peer review and any attached files.

Reviewer #1: **Yes: **Tanguy Fardet

Reviewer #2: No

Reviewer #3: No

**Figure resubmission:**
---

## [Decision Letter · Decision Letter 3]

20 May 2025

Dear Mr. Jaras,

We are pleased to inform you that your manuscript 'Unveiling the role of local metabolic constraints on the structure and activity of spiking neural networks' has been provisionally accepted for publication in PLOS Computational Biology.

Best regards,

Fleur Zeldenrust

Academic Editor

PLOS Computational Biology

Marieke van Vugt

Section Editor

PLOS Computational Biology

I believe there is an interesting discussion between reviewer 2 and the authors, that nevertheless goes a bit beyond the scope of this manuscript. Therefore, I propose to publish the manuscript in its current form, and hopefully the authors and reviewer see this discussion of a starting point of some further work on this topic.

Reviewer's Responses to Questions

**Comments to the Authors:**

Reviewer #2: Response to Comment 6.

I’m afraid the authors’ responses missed my points. First, yes, I see that everything saturated. The point was exactly that - the excitatory rates saturate not because of any inhibitory inputs but because the excitatory weights/refractory mechanisms hit a bound. As far as I can tell, the main effect of inhibition presented in the paper is to capture and silence some E neurons in fig. 6. The response provided argued that inhibition helps stabilize the network in a metabolic crisis - but then again, I would argue that the core network dynamics are first governed by metabolism in this case, since, as shown by fig. 5, without energy constraints, other non-inhibitory mechanisms cause E saturation. Yet, I will acknowledge that adjusting the dynamical regime would be too much to ask. Rather, upon further reflection, I think the first paragraph in the ‘Network Model’ section is misleading, by still mentioning “balanced E-I networks”, because it hints at a dynamical regime which is not explored. I don’t think anyone would argue the utility of E/I networks, so I would simply remove this paragraph; at the very least, remove “balanced”.

Second, I was not at all suggesting to incorporate memory mechanisms here; rather, it was a rebuttal to the thought experiment proposed in the previous Comment 25 regarding constant inputs to E and I inducing a weakening of weights. The point was that, if you put this thought experiment into a plausible biological context, it appears to be problematic. Again, yes, nothing to address here, but perhaps to keep in mind for future work.

No further comments.

**Have the authors made all data and (if applicable) computational code underlying the findings in their manuscript fully available?**

Reviewer #2: None

PLOS authors have the option to publish the peer review history of their article (what does this mean?). If published, this will include your full peer review and any attached files.

Reviewer #2: No

---

## [Editor Report · Acceptance letter]

PCOMPBIOL-D-23-01765R3

Unveiling the role of local metabolic constraints on the structure and activity of spiking neural networks

Dear Dr Jaras,

I am pleased to inform you that your manuscript has been formally accepted for publication in PLOS Computational Biology. Your manuscript is now with our production department and you will be notified of the publication date in due course.

With kind regards,

Zsofia Freund
